# MoSSDA: A Semi-Supervised Domain Adaptation Framework for Multivariate Time-Series Classification using Momentum Encoder

## Abstract

Deep learning has emerged as the most promising approach in various fields; however, when the distributions of training and test data are different (domain shift), the performance of deep learning models can degrade. Semi-supervised domain adaptation (SSDA) is a major approach for addressing this issue, assuming that a fully labeled training set (source domain) is available, but the test set (target domain) provides labels only for a small subset. In this study, we propose a novel two-step momentum encoder-utilized SSDA framework, MoSSDA, for multivariate time-series classification. Time series data are highly sensitive to noise, and sequential dependencies cause domain shifts resulting in critical performance degradation. To obtain a robust, domain-invariant and class-discriminative representation, MoSSDA employs a domain-invariant encoder to learn features from both source and target domains. Subsequently, the learned features are fed to a mixup-enhanced positive contrastive module consisting of an online momentum encoder. The final classifier is trained with learned features that exhibit consistency and discriminability with limited labeled target domain data. We applied a two-stage process by separating the gradient flow between the encoders and the classifier to obtain rich and complex representations. Through extensive experiments on six diverse datasets, MoSSDA achieved state-of-the-art performance for three different backbones and various unlabeled ratios in the target domain data. The Ablation study confirms that each module, including two-stage learning, is effective in improving the performance.

## 1 Introduction

The advent of deep learning has led numerous models demonstrating remarkable performance across various domains. Specifically, time-series classification has become a significant and challenging problem in various applications, including medicine, manufacturing, and human activity recognition Eldele et al. (2021a); Chang et al. (2020); Li et al. (2021c); Ragab et al. (2023); Deng et al. (2021). Time-series data require a different approach compared to other data types because their continuous nature includes temporal dependencies, trends, and recurring patterns. In the case of multivariate time series, the data becomes even more complex because of the intermingling of channel dynamics and channel-dependency information. These inherent characteristics make multivariate time-series classification particularly challenging.

Real-world time-series data are prone to variations owing to factors, such as collection environment, sensor type, and recording conditions. Therefore, time-series data often exhibit significant shifts in their distribution. This phenomenon, termed "domain shift," violates the fundamental independent and identically distributed (i.i.d.) assumption underlying numerous machine learning models Ott et al. (2022). However, deep-learning-based time-series models tend to degrade their performance when the test data distribution (target domain) differ from that of the training data (source domain). In practice, this assumption is frequently violated: in the medical domain, biosignals such as ECG and PPG can differ across hospitals owing to variations in acquisition devices and preprocessing pipelines, leading to distinct noise, amplitude, and frequency characteristics Wagner et al. (2022); in industrial process monitoring, changes in materials equipment, or operating conditions can alter the signatures of the same fault type Lessmeier et al. (2016); and in smartwatch-based sensing,

user-specific factors (e.g., body shape, wearing habits, daily activity patterns) induce user-dependent signal distributions Anguita et al. (2013); Stisen et al. (2015); Kwapisz et al. (2011). Consequently, a model $F$ trained to potimal on a source distribution $f$ trained to be optimal on a source distribution $p_s(x, y)$ tends to specialize in source-specific statistical regularities; under a different target distribution $p_t(x, y)$, these regularities may no longer hold or even become misleading, resulting in a noticeable degradation in generalization performance.

The domain adaptation approach aims to address this challenge and has gained interest from deep learning and time-series researchers. Two main research directions exist in domain adaptation: unsupervised and semi-supervised domain adaptation (SSDA). Unsupervised domain adaptation (UDA) methods presume the complete absence of labels in the target domain. However, many practical scenarios allow for the acquisition of a limited valuable set of labeled target domain data. In such settings, which are addressed by SSDA, a pragmatic approach is to leverage a small set of labeled target instances alongside a larger corpus of unlabeled data. This strategy was demonstrated to be effective in resolving the distribution discrepancy between the source and target domains Saito et al. (2019); Kim et al. (2022).

Data augmentation has emerged as a prominent strategy for utilizing limited target domain information, largely motivated by its profound success in computer vision Ilbert et al. (2024); Iglesias et al. (2023). However, because temporal order and sequential dependencies are crucial in time-series data, the common transformation-based augmentation used in spatial data, such as rotation or random cropping, may disrupt critical temporal characteristics Chang et al. (2024) and degrade model performance.

In this study, we propose a novel two-step **Mo**mentum encoder-utilized **SSDA** framework, **MoSSDA**, for multivariate time-series classification. Generic SSDA typically trains encoder and classifier jointly with a single combined loss and relies on input-space augmentations or Mixup. By contrast, our method (a) separates representation learning from classifier training to mitigate gradient conflicts that are particularly severe under temporal noise, (b) avoids manipulating the raw temporal axis and performs Mixup only after temporal encoding, and (c) uses a momentum encoder to stabilize cross-domain temporal patterns when strong input augmentations are intentionally avoided. We evaluated our method using benchmark real-world, four different multivariate time series datasets Kwapisz et al. (2011); Stisen et al. (2015); Anguita et al. (2013); Wagner et al. (2022), and two different univariate time series datasets Lessmeier et al. (2016); Goldberger et al. (2000), and achieved state-of-the-art performances.

The key contributions of this study are summarized as follows:

- We propose a novel SSDA strategy that emphasizes the extraction of robust, domain-invariant, and inter- and intra-domain class-discriminative representations, using three different modules and a decoupled two-step training process.

- A domain-invariant encoder that mitigates domain shifts by utilizing both unlabeled and labeled data, was employed using the maximum mean discrepancy (MMD) loss to learn domain-invariant features.

- In the positive contrastive module, inter- and intra-domain representations are learned through supervised contrastive loss without input-level time-series augmentations that disrupt temporal structure, instead employing feature-space mixup to preserve temporal dependencies while enhancing class discriminability. Mixup allows MoSSDA to efficiently utilize limited target domain data as rich representations without input-level augmentations process commonly used in prior SSDA methods.

- A momentum encoder was employed to enhance feature consistency and discriminability by preserving semantically meaningful representations across iterations.

- Through extensive experiments on time-series classification, we demonstrated that the proposed method achieves state-of-the-art performance, significantly outperforming existing widely used SSDA approaches.

## 2 RELATED WORK

### 2.1 TIME SERIES DOMAIN ADAPTATION

The objective of time-series domain adaptation is to learn feature representations that are domain-invariant (ensuring robustness across domains) and class-discriminative (ensuring classification effectiveness) Ott et al. (2022); Shi et al. (2022); Chen et al. (2024); Liu & Xue (2021). To meet this requirement, specialized methods have been developed to preserve the unique temporal characteristics of the data. Most approaches have focused on the alignment of feature distributions across domains. A prominent strategy involves minimizing statistical distance metrics, such as MMD in a Reproducing Kernel Hilbert Space (RKHS) Ott et al. (2022). AdvSKM Liu & Xue (2021) was proposed with a novel spectral kernel to enhance the MMD metric for more accurate discrepancy measurements. In another study, MMD was combined with feature transformation techniques, such as correlation alignment, to map source features more closely to the target distribution He et al. (2023). Other significant research efforts have utilized adversarial training. Adversarial-training-based methods comprise a domain discriminator that distinguishes features from the source and target domains, whereas the feature extractor is trained to generate domain-agnostic features Wilson et al. (2020). For instance, DAF Jin et al. (2022) incorporates a domain discriminator with shared attention modules for time-series forecasting. Recent methods, such as CADT Chen et al. (2024), focus on disentangling domain-invariant features from domain-specific one, and use custom contrastive learning objectives to address the instabilities common in adversarial architectures. Contrastive learning has recently emerged as an effective technique for domain alignment in time-series studies. CoTMix Eldele et al. (2023) is an example that exclusively utilizes contrastive objectives to mitigate distribution shifts. DACAD Darban et al. (2024) integrates contrastive learning with the UDA for anomaly detection by incorporating supervised and self-supervised contrastive losses into the source and target domains respectively. In addition to these alignment- and contrastive-based approaches, recent work has explored more structured mechanisms for adaptation. SASA2 Li et al. (2024) leverages sparse associative structure alignment, aligning domain-specific associative patterns to cpature relational invariances across domains better. POND Wang et al. (2024) adopts prompt-based adaptation for time series, where learnable prompts or domain tokens are tuned—rather than the entire backbone—to steer pre-trained models toward target domains, as in prompt-based domain discrimination frameworks for multi-source time-series adaptation.

### 2.2 SEMI-SUPERVISED DOMAIN ADAPTATION

In the SSDA framework, a limited number of labeled samples are available in the target domain. These samples can be utilized in conjunction with a substantial unlabeled data corpus to significantly enhance performance Yoon et al. (2022); Cheng & Pan (2014). A prevalent technique in SSDA involves the application of consistency regularization, which is frequently accompanied by data augmentation. AdaMatch Berthelot et al. (2021) enforces consistency between the predictions on weakly and strongly augmented versions of the target samples to align the distributions. Similarly, DARK Kim et al. (2022) employs cross-view consistency regularization to distill domain-specific knowledge. Another study demonstrated a combining self-supervised pretraining with consistency regularization can yield strong results without explicit domain alignment Mishra et al. (2021). Pseudo-labeling is another prevalent SSDA method. This method leverages the model's high-confidence predictions of unlabeled target data as "pseudo-labels" to expand the training set. DECOTA Yang et al. (2021) employs a co-training framework that utilizes pseudo-labels to decompose the SSDA tasks. However, naïve pseudo-labeling can reinforce confirmation bias. Methods, such as UniSSDA Zhang et al. (2024), have proposed prior-guided refinement strategies to mitigate this issue, particularly in challenging settings with private classes. Other approaches have focused on adversarial training and clustering. For instance, a minimax entropy approach was proposed to adversarially optimize a few-shot model Saito et al. (2019), whereas CDAC Li et al. (2021a) utilizes adversarial adaptive clustering loss to align inter- and intra-domain distributions. Various prevailing methodologies depend on sophisticated data augmentation strategies Kim et al. (2022); Berthelot et al. (2021); Li et al. (2021b) or intricate end-to-end adversarial training Ganin & Lempitsky (2015); Long et al. (2018); Shu et al. (2018).

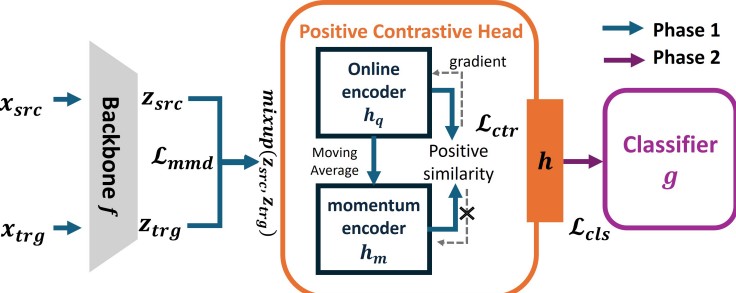

Figure 1: **Overview of MoSSDA framework.** In the decoupled two-step framework, the first step involves updating the components (colored in navy): The following part is used to update the classifier (colored in purple). The weights in the other steps were not updated for each step.

## 3 METHODOLOGY

### 3.1 PROBLEM FORMULATION

In the SSDA setting for multivariate time-series classification, source and target domain datasets were provided. The source domain dataset is fully labeled, while the target domain dataset contains only a few labeled samples, with the remainder unlabeled. The source domain dataset is expressed as follows: $\mathcal{D}_{\text{src}} = \{(X_{\text{src}}^i, y_{\text{src}}^i)\}_{i=1}^{N_s}$, where $N_s$ is the number of source samples. Each sample, $X_{\text{src}}^i \in \mathbb{R}^{D \times T}$, is a multivariate time-series instance with $D$ variables (channels) and $T$ time steps, and $y_{\text{src}}^i \in \mathcal{Y}$ is its corresponding class label. The target domain dataset, $\mathcal{D}_{\text{trg}}$, consisted of two distinct subsets. A small labeled set, $\mathcal{D}_{\text{trg}}^\ell = \{(X_{\text{trg}}^j, y_{\text{trg}}^j)\}_{j=1}^{N_t^\ell}$, where $N_t^\ell$ is the number of labeled target samples, and a large unlabeled set, $\mathcal{D}_{\text{trg}}^u = \{X_{\text{trg}}^k\}_{k=1}^{N_t^u}$, where $N_t^u$ is the number of unlabeled target samples. Naturally, the number of labeled target data is smaller than that of unlabeled ones, i.e., $N_t^\ell \ll N_t^u$. We denote the combined set of all available labeled data as $\mathcal{D}^\ell = \mathcal{D}_{\text{src}} \cup \mathcal{D}_{\text{trg}}^\ell$. We assume that the source and target domains share the same label space $\mathcal{Y}$ but their data distributions $P(X_{\text{src}})$ and $P(X_{\text{trg}})$ can be different. The main objective of our framework is to minimize the domain discrepancy between the source and target domains, and to learn a task-specific classifier using $\mathcal{D}_{\text{src}}$ and $\mathcal{D}_{\text{trg}}$ to accurately predict labels on test data from the target domain.

### 3.2 METHOD DESCRIPTION

The framework of the proposed **MoSSDA** for multivariate time-series semi-supervised domain adaptation is shown in Figure 1. In the first stage, the framework utilizes the full spectrum of available data, labeled source domain data $\mathcal{D}_{\text{src}}$, labeled target domain data $\mathcal{D}_{\text{trg}}^\ell$, and unlabeled target domain data $\mathcal{D}_{\text{trg}}^u$, to learn feature representations. The learned feature representations are simultaneously domain-invariant and class-discriminative, promoting both intra- and inter-domain class separability by employing **a Domain-Invariant Encoder** and **Positive Contrastive Module**. The second stage trains a high-performance **Classification Module** on the learned feature representation using only the reliable labeled data $\mathcal{D}^\ell$. This separation prevents conflicting optimization objectives and enhances training stability. In summary, the important novelty of the MoSSDA is the explicit separation of representation learning (stage 1) and classifier training (stage 2), which eliminates conflicting gradients—unlike previous works that jointly optimize domain alignment and class discrimination, often resulting in training instability. This two-step approach leverages the full spectrum of available data to first learn domain-invariant and class-discriminative representations, and then trains the classifier solely on reliable labeled data, ensuring stable and robust adaptation performance.

### 3.3 DOMAIN-INVARIANT ENCODER

The first component of a framework is the domain-invariant encoder, which is designed to learn feature representations that are robust to distributional shifts between the source and target domains. We denote this feature extractor as a network $f$, parameterized by $\theta_f$. The encoder maps an input

time-series $X$ to a latent representation $Z = f(X)$. Let $Z_{\text{src}} = f(X_{\text{src}})$ and $Z_{\text{trg}} = f(X_{\text{trg}}) = \{Z_{\text{trg}}^{\ell}, Z_{\text{trg}}^{u}\}$ be the sets of feature representations for the source and target domains, respectively.

To obtain domain invariance, we employed MMD loss, which is a widely used metric for comparing distributions in RKHS. The squared MMD between the source and target feature distributions was used as the MMD loss, which can be computed over batches as follows:

$$\mathcal{L}_{\text{mmd}} = \left\| \mathbb{E}_{z_s \sim Z_{\text{src}}}[\phi(z_s)] - \mathbb{E}_{z_t \sim Z_{\text{trg}}}[\phi(z_t)] \right\|_{\mathcal{H}}^{2} \tag{1}$$

where $z_s \in Z_{\text{src}}$, $z_t \in Z_{\text{trg}}$, and $\phi(\cdot)$ is a mapping to the RKHS $\mathcal{H}$. The empirical estimate of this loss, using the kernel trick $k(x, x') = \langle \phi(x), \phi(x') \rangle$ In this work, we primarily used a linear kernel, $k(x, x') = x^{\top} x'$, for its simplicity and computational efficiency. However, our framework is flexible, allowing for the substitution of other kernels (e.g., RBF) as required by the specific dataset characteristics. In our appendix, A.7.2 presents the result of various kernels for MMD loss.

### 3.4 POSITIVE CONTRASTIVE MODULE

The second key objective of the framework is to learn feature representations that are class-discriminative both within each domain (intra-domain) and across them (inter-domain). This can be obtained by leveraging the labeled data from both domains, $\mathcal{D}^{\ell}$. A significant challenge arises from scarcity of labeled target data, which can lead to the feature space to be highly biased towards the source domain. To compensate for this bias and promote robust feature learning, we designed a strategy that combines mixup Zhang et al. (2017) with a supervised contrastive loss Khosla et al. (2020); Grill et al. (2020). Traditional input-level augmentations (e.g., rotation, cropping, permutation, random erasing, time warping) can fundamentally alter or destroy the temporal ordering and dependencies that are semantically meaningful in time series. The feature-space mixup operates in the learned representation space where temporal structure has already been implicitly encoded by the encoder, thus avoiding direct manipulation of temporal relationships.

We created a positive counterpart of $z_i$ by linearly interpolating its feature representation with that of another sample ($z_j$) from the **same class**, which may belong to the source or target domain, as follows:

$$z_{\text{mix}} = \lambda z_i + (1 - \lambda) z_j \tag{2}$$

where $y_i = y_j, \lambda \sim \text{Beta}(\alpha, \alpha)$. This process enables the model to learn smoother decision boundaries and creates a more diverse set of positive examples for the subsequent contrastive learning steps, without input-level time-series augmentations that disrupt temporal structure. The set of latent representations used for this module is denoted $\mathcal{Z}^{\ell} = \{Z_{\text{src}}, Z_{\text{trg}}^{\ell}, Z_{\text{mix}}\}$, thus comprises original labeled features from both domains and their mixed-up counterparts. With the enriched labeled set $Z^{\ell}$, we train the encoder using a supervised contrastive loss Khosla et al. (2020); Grill et al. (2020); Chen & He (2021).

The positive supervised contrastive loss function encourages feature representations of samples from the same classes (positive) should be closer together in the embedding space. For a training batch, the loss can be defined as:

$$\mathcal{L}_{ctr} = \mathbb{E}\left[ -\log \frac{\sum_{j \neq i} I\left(y_i = y_j\right) \exp\left(\text{sim}\left(z_i, z_j\right)/\tau\right)}{\sum_{k \neq i} \exp\left(\text{sim}\left(z_i, z_k\right)/\tau\right)} \right] \tag{3}$$

where $\tau$ is a temperature hyperparameter, and $\text{sim}(u, v) = u^{\top} v/(\|u\|\|v\|)$ denotes the cosine similarity. The objective of this loss function is to leverage class labels to capture the embedding space and encourage the samples to belong to the same class—regardless of their domain of origin—to formed compact and well-separated clusters. Specifically, positive pairs are constructed using a combination of samples from the source domain, target domain, and their interpolated mixtures via a mixup. This design integrates both intra- and inter-domain positive pairs into the learning objective, thereby encouraging the acquisition of class-discriminative and domain-robust representations.

To ensure that the feature representations used for contrastive learning were stable and consistent, we employed the momentum encoder pioneered by MoCo He et al. (2020). Thus, the positive contrastive module comprises two encoders.

- **An online encoder**: $h_q$ parameterized by weights $\theta_q$. These weights are actively updated via backpropagation from the learning objectives.

- **A momentum encoder**: $h_m$ parameterized by weights $\theta_m$. These weights were **not** updated via backpropagation.

Instead of direct gradient updates, the momentum encoder's weights, $\theta_m$, are updated as an *exponentially moving average* (EMA) of the online encoder's weights, $\theta_q$. After each training step, an update was performed as follows:

$$\theta_m \leftarrow m \cdot \theta_m + (1 - m) \cdot \theta_q \tag{4}$$

where $m \in [0, 1)$ is the momentum coefficient, a hyperparameter that controls the speed of the update. $\theta_q$ represents the weight of the online encoder after the gradient update in the current training step. $\theta_m$ represents the weights of the momentum encoder, which are being updated by Eq. 4. The momentum coefficient $m$ is typically set to a large value, such as 0.999. This is highly beneficial for contrastive learning because it provides a stable and consistent feature representation. This prevents instability in the learning process, which can occur if the feature keys in the contrastive dictionary change rapidly at every gradient step. Consequently, the model can effectively learn robust time-series features without requiring a large labeled training batch.

## 3.5 CLASSIFICATION MODULE

In the second stage, we train a high-performance classifier using this enhanced labeled set. All the modules learned in the first stage were frozen, and the classifier was trained solely on the optimized features extracted from the frozen encoder.

The classification loss employed is the standard cross-entropy loss, defined as:

$$\mathcal{L}_{\text{ce}} = -\sum_{i=1}^{N} y_i \log\left(\widehat{y}_i\right) \tag{5}$$

where $y_i$ denotes the ground truth label and $\widehat{y}_i$ the predicted probability of the $i$-th sample. The final classification loss can be defined as:

$$\mathcal{L}_{\text{cls}} = \mathcal{L}_{\text{ce}}^{s} + \mathcal{L}_{\text{ce}}^{t} \tag{6}$$

where $\mathcal{L}_{\text{ce}}^{s}$ and $\mathcal{L}_{\text{ce}}^{t}$ depict the cross-entropy losses on the source and target domains, respectively. This loss function encourages the classifier to learn class-discriminative decision boundaries, thereby improving its ability to correctly classify samples from both source and target domains based on their respective labels.

## 3.6 OVERALL LOSS

The overall training objective consisting of the MMD loss, positive supervised contrastive loss, and classification loss can be formulated as Eq. 7 and 8

Specifically, the overall loss for the first stage is a weighted sum of the MMD loss and the supervised contrastive loss, as follows:

$$\mathcal{L}_{\text{stage1}} = \lambda_{\text{mmd}} * \mathcal{L}_{\text{mmd}} + \lambda_{\text{ctr}} * \mathcal{L}_{\text{ctr}} \tag{7}$$

where $\lambda_{\text{mmd}}$ and $\lambda_{\text{ctr}}$ are hyperparameters that balance the contribution of each loss term. The loss in the second stage minimizes the standard cross-entropy loss over all available reliably labeled data.

$$\mathcal{L}_{\text{stage2}} = \mathcal{L}_{\text{ce}}(\mathcal{D}^{\ell}) = \mathcal{L}_{\text{ce}}^{s} + \mathcal{L}_{\text{ce}}^{t} \tag{8}$$

Simultaneous training of both stages may cause the first stage to create a simple latent representation that can be easily classified in the second stage, potentially leading to an overfitting. Hence, we train the first stage using Eq. 7 to construct a robust, domain-agnostic feature space and, then train the second stage with Eq. 8 to learn the optimal decision boundaries within space.

## 4 EXPERIMENTS

### 4.1 EXPERIMENTAL SETUP

**Datasets.** The experiment encompassed six time-series datasets from diverse domains, four multi-variate time-series datasets, and two univariate time-series datasets, namely, UCIHAR Anguita et al. (2013), HHAR Stisen et al. (2015), WISDM Kwapisz et al. (2011) EEG Goldberger et al. (2000), PTBXL Wagner et al. (2020; 2022), and MFD Lessmeier et al. (2016). These datasets are commonly used for time-series domain adaptation tasks, except for the PTBXL dataset. A detailed description of the datasets is provided in the Appendix A.1.

**Backbones.** Similar to previous studies on time-series domain adaptation Ragab et al. (2023); Sun et al. (2024); Chen et al. (2024), we employed ResNet18 He et al. (2016); Fawaz (2020), a CNN Eldele et al. (2021b; 2022), and a TCN Bai et al. (2018); Thill et al. (2020) as the backbone networks in our experiments. A 1D-CNN utilizes three convolutional blocks, each comprising a 1D-convolutional layer, BatchNorm, ReLU activation, and MaxPooling. RESNET18 implements a 1D residual network with shortcut connections between successive convolutional layers to enable deeper architectures. A TCN employs causal dilated convolutions to prevent temporal information leakage while capturing long-range dependencies in time-series data.

**Adaptation Scenarios.** For a fair comparison, we used the same setting for the benchmark datasets as in the prior work Ragab et al. (2023), including the data-splits and adaptation scenarios. In the case of PTBXL, we employed all six combinations of domains. We used a consistent setting across all experiments.

**Benchmark Methods.** State-of-the-art SSDA methods, CDAC Li et al. (2021a), PAC Mishra et al. (2021), AdaMatch Berthelot et al. (2021), and UniSSDA Zhang et al. (2024), were employed for comparison. In addition, we employed DST Chen et al. (2022) and contrastive learning-based SSDA (CLDA) Singh (2021). All the benchmark methods employed augmentation techniques, implemented by adapting image-specific augmentations to time-series-specific augmentations.

**Implementation.** Each minibatch of size $B$ of source and target domain samples is equal, while the target domain samples consist of unlabeled data and labeled data, as provided in the unlabeled ratios: $u \in \{0.7, 0.9, 0.95\}$ across the entire experiments. We set hyperparameter temperature $\tau = 0.5$, momentum coefficient $m = 0.999$, $\alpha = 1$, and both $\lambda_{\mathrm{mmd}}$ and $\lambda_{\mathrm{ctr}}$ set to 0.5. All the experiments were performed using PyTorch with an NVIDIA RTX 6000 Ada Generation system. The implementation details are described in Appendix A.3, and the experimental results of different hyperparameter settings are compared in Appendix A.7.1.

### 4.2 EXPERIMENTAL RESULTS

#### 4.2.1 PERFORMANCE COMPARISON.

| | Averaged rank of averaged test accuracy | | | | | | |
|---|---|---|---|---|---|---|---|
| $u$ | AdaMatch | CDAC | DST | PAC | UniSSDA | CLDA | **OURS** |
| 0.7 | 3.94 | 6.94 | 3.56 | 3.50 | 4.11 | 4.83 | 1.11 |
| 0.9 | 3.67 | 6.89 | 3.17 | 4.11 | 4.28 | 4.83 | 1.06 |
| 0.95 | 3.47 | 6.88 | 3.53 | 3.88 | 4.41 | 4.76 | 1.06 |

Table 1: Averaged rank on overall results with SSDA methods.

Table 1 summarizes the results from Tables 2, S2, and S3 by referring to the average ranking of each method across all experimental settings. A lower rank indicates better performance. MoSSDA achieved the best overall ranking across all unlabeled ratio conditions, consistently outperforming the existing benchmark methods. Other semi-supervised methods varied depending on the amount of labeled data; however, MoSSDA retained its robustness. In the context of the relatively generous condition (unlabeled ratio = 0.7), PAC demonstrated the second-best performance, which can

| unlbl ratio | metric | AdaMatch Avg. | std. | CDAC Avg. | std. | DST Avg. | std. | PAC Avg. | std. | UniSSDA Avg. | std. | CLDA Avg. | std. | OURS Avg. | std. |
|---|---|---|---|---|---|---|---|---|---|---|---|---|---|---|---|
| **EEG** | | | | | | | | | | | | | | | |
| 0.7 | acc | 0.5107 | ±0.11 | 0.1282 | ±0.06 | 0.5045 | ±0.12 | 0.6144 | ±0.12 | 0.4527 | ±0.09 | 0.3999 | ±0.16 | **0.7910** | ±0.05 |
| | f1 | 0.3837 | ±0.11 | 0.0478 | ±0.03 | 0.3885 | ±0.11 | 0.5029 | ±0.13 | 0.3455 | ±0.10 | 0.3257 | ±0.15 | **0.6862** | ±0.05 |
| 0.9 | acc | 0.4767 | ±0.11 | 0.1282 | ±0.06 | 0.4609 | ±0.13 | 0.4835 | ±0.12 | 0.4160 | ±0.10 | 0.3184 | ±0.13 | **0.7553** | ±0.07 |
| | f1 | 0.3816 | ±0.10 | 0.0479 | ±0.03 | 0.3776 | ±0.12 | 0.3545 | ±0.13 | 0.3223 | ±0.10 | 0.2603 | ±0.12 | **0.6552** | ±0.06 |
| 0.95 | acc | 0.4684 | ±0.11 | 0.1286 | ±0.07 | 0.4558 | ±0.12 | 0.4503 | ±0.16 | 0.4046 | ±0.09 | 0.3147 | ±0.12 | **0.7328** | ±0.07 |
| | f1 | 0.3806 | ±0.11 | 0.0480 | ±0.03 | 0.3806 | ±0.11 | 0.3064 | ±0.13 | 0.3121 | ±0.10 | 0.2616 | ±0.12 | **0.6244** | ±0.06 |
| **HAR** | | | | | | | | | | | | | | | |
| 0.7 | acc | 0.6097 | ±0.10 | 0.1524 | ±0.02 | 0.5931 | ±0.06 | 0.4773 | ±0.20 | 0.5465 | ±0.08 | 0.2639 | ±0.13 | **0.9594** | ±0.04 |
| | f1 | 0.5284 | ±0.11 | 0.0496 | ±0.01 | 0.5056 | ±0.07 | 0.3972 | ±0.24 | 0.4319 | ±0.09 | 0.1884 | ±0.13 | **0.9606** | ±0.04 |
| 0.9 | acc | 0.6001 | ±0.09 | 0.1479 | ±0.02 | 0.5880 | ±0.06 | 0.2777 | ±0.08 | 0.5191 | ±0.04 | 0.2665 | ±0.12 | **0.9376** | ±0.05 |
| | f1 | 0.5333 | ±0.11 | 0.0461 | ±0.01 | 0.4946 | ±0.07 | 0.1682 | ±0.07 | 0.3862 | ±0.04 | 0.1881 | ±0.14 | **0.9392** | ±0.04 |
| 0.95 | acc | 0.6071 | ±0.09 | 0.1501 | ±0.02 | 0.5871 | ±0.07 | 0.2889 | ±0.12 | 0.5221 | ±0.04 | 0.2660 | ±0.13 | **0.8970** | ±0.10 |
| | f1 | 0.5332 | ±0.11 | 0.0484 | ±0.01 | 0.4912 | ±0.08 | 0.1656 | ±0.10 | 0.3956 | ±0.05 | 0.1902 | ±0.14 | **0.8948** | ±0.10 |
| **HHAR** | | | | | | | | | | | | | | | |
| 0.7 | acc | 0.5130 | ±0.16 | 0.1651 | ±0.02 | 0.5150 | ±0.18 | 0.5449 | ±0.16 | 0.5107 | ±0.16 | 0.4897 | ±0.17 | **0.9693** | ±0.02 |
| | f1 | 0.4949 | ±0.18 | 0.0532 | ±0.02 | 0.4987 | ±0.21 | 0.4954 | ±0.17 | 0.4914 | ±0.19 | 0.4536 | ±0.18 | **0.9698** | ±0.02 |
| 0.9 | acc | 0.5093 | ±0.16 | 0.1654 | ±0.02 | 0.5159 | ±0.18 | 0.4228 | ±0.17 | 0.4987 | ±0.16 | 0.4632 | ±0.16 | **0.9563** | ±0.02 |
| | f1 | 0.4935 | ±0.18 | 0.0535 | ±0.02 | 0.5071 | ±0.21 | 0.3587 | ±0.20 | 0.4830 | ±0.18 | 0.4293 | ±0.18 | **0.9567** | ±0.02 |
| 0.95 | acc | 0.5096 | ±0.16 | 0.1651 | ±0.02 | 0.5043 | ±0.18 | 0.2822 | ±0.14 | 0.4998 | ±0.16 | 0.4615 | ±0.16 | **0.9430** | ±0.03 |
| | f1 | 0.4915 | ±0.18 | 0.0532 | ±0.02 | 0.4900 | ±0.21 | 0.1817 | ±0.11 | 0.4816 | ±0.18 | 0.4238 | ±0.17 | **0.9433** | ±0.03 |
| **MFD** | | | | | | | | | | | | | | | |
| 0.7 | acc | 0.6990 | ±0.26 | 0.4550 | ±0.00 | 0.5805 | ±0.10 | 0.8036 | ±0.09 | 0.6844 | ±0.26 | 0.6910 | ±0.12 | **0.9726** | ±0.04 |
| | f1 | 0.6868 | ±0.30 | 0.2085 | ±0.00 | 0.6216 | ±0.18 | 0.7937 | ±0.17 | 0.6739 | ±0.30 | 0.7096 | ±0.14 | **0.9571** | ±0.07 |
| 0.9 | acc | 0.6932 | ±0.27 | 0.4550 | ±0.00 | 0.6949 | ±0.27 | 0.3772 | ±0.10 | 0.6801 | ±0.25 | 0.6788 | ±0.19 | **0.9339** | ±0.11 |
| | f1 | 0.6811 | ±0.30 | 0.2085 | ±0.00 | 0.6803 | ±0.32 | 0.2751 | ±0.11 | 0.6709 | ±0.30 | 0.6898 | ±0.20 | **0.9065** | ±0.15 |
| 0.95 | acc | 0.6907 | ±0.27 | 0.4550 | ±0.00 | 0.5622 | ±0.05 | 0.6584 | ±0.08 | 0.6801 | ±0.26 | 0.6764 | ±0.20 | **0.9519** | ±0.06 |
| | f1 | 0.6784 | ±0.30 | 0.2085 | ±0.00 | 0.5849 | ±0.06 | 0.6444 | ±0.11 | 0.6720 | ±0.30 | 0.6858 | ±0.21 | **0.9096** | ±0.14 |
| **PTBXL** | | | | | | | | | | | | | | | |
| 0.7 | acc | 0.4816 | ±0.07 | 0.2197 | ±0.17 | 0.4521 | ±0.05 | 0.4869 | ±0.13 | 0.4867 | ±0.08 | 0.4835 | ±0.09 | **0.7361** | ±0.04 |
| | f1 | 0.2720 | ±0.04 | 0.0934 | ±0.03 | 0.2585 | ±0.07 | 0.3939 | ±0.06 | 0.2157 | ±0.07 | 0.2254 | ±0.05 | **0.6179** | ±0.07 |
| 0.9 | acc | 0.4402 | ±0.06 | 0.2236 | ±0.17 | 0.4044 | ±0.06 | 0.5359 | ±0.04 | 0.4350 | ±0.07 | 0.4380 | ±0.07 | **0.7213** | ±0.02 |
| | f1 | 0.2700 | ±0.03 | 0.0953 | ±0.03 | 0.2620 | ±0.07 | 0.2573 | ±0.03 | 0.2162 | ±0.07 | 0.2667 | ±0.05 | **0.5880** | ±0.04 |
| 0.95 | acc | 0.4302 | ±0.06 | 0.2239 | ±0.17 | 0.4013 | ±0.06 | 0.5167 | ±0.10 | 0.4064 | ±0.08 | 0.3791 | ±0.10 | **0.7014** | ±0.01 |
| | f1 | 0.2831 | ±0.02 | 0.0958 | ±0.03 | 0.2715 | ±0.06 | 0.2144 | ±0.05 | 0.2098 | ±0.07 | 0.2726 | ±0.05 | **0.5701** | ±0.05 |
| **WISDM** | | | | | | | | | | | | | | | |
| 0.7 | acc | 0.3358 | ±0.12 | 0.0897 | ±0.04 | 0.5356 | ±0.19 | 0.5035 | ±0.18 | 0.3486 | ±0.12 | 0.1352 | ±0.09 | **0.7838** | ±0.04 |
| | f1 | 0.0996 | ±0.05 | 0.0450 | ±0.04 | 0.3503 | ±0.16 | 0.2018 | ±0.10 | 0.1817 | ±0.10 | 0.1006 | ±0.08 | **0.7176** | ±0.08 |
| 0.9 | acc | 0.2419 | ±0.14 | 0.0883 | ±0.04 | 0.5091 | ±0.19 | 0.3548 | ±0.16 | 0.3697 | ±0.11 | 0.1305 | ±0.08 | **0.7403** | ±0.11 |
| | f1 | 0.0657 | ±0.03 | 0.0399 | ±0.03 | 0.3599 | ±0.18 | 0.1222 | ±0.08 | 0.1831 | ±0.07 | 0.0998 | ±0.07 | **0.6709** | ±0.15 |
| 0.95 | acc | 0.2079 | ±0.15 | 0.0966 | ±0.05 | 0.4991 | ±0.19 | 0.3389 | ±0.16 | 0.3766 | ±0.12 | 0.1353 | ±0.10 | **0.6732** | ±0.13 |
| | f1 | 0.0703 | ±0.03 | 0.0499 | ±0.04 | 0.3584 | ±0.16 | 0.1225 | ±0.07 | 0.2093 | ±0.11 | 0.1021 | ±0.08 | **0.6084** | ±0.16 |

Table 2: **Comparison with SSDA methods**: Averaged target domain test accuracy and f1-score across domain pairs for each datasets with RESNET18 backbone. The best performance is in bold and the second best is underlined.

be attributed to its utilization of pretraining and consistency regularization. However, under more challenging conditions (unlabeled ratio = 0.9 and 0.95), DST and AdaMatch become the strongest baselines. MoSSDA surpasses these methods that rely heavily on pseudo-labeling or generic augmentations by more effectively exploiting the limited label information in the target domain.

The proposed method was evaluated using three widely adopted backbone architectures, and its performance was compared with those of six state-of-the-art domain adaptation benchmark methods. The evaluation metrics included the mean accuracy and F1-score under the domain adaptation scenarios. Table 2 presents the results obtained using RESNET18 as the underlying framework across six time series datasets with three distinct unlabeled ratios.

The proposed method (MoSSDA) exhibited consistent superiority over other benchmark methods in target domain classification, particularly with the RESNET18. PAC demonstrated a competitive performance following MoSSDA; however, it experienced a substantial decrease in F1-score under more challenging settings (unlabeled ratio = 0.95). In contrast, MoSSDA demonstrated consistent performance, even when managing class-imbalanced datasets, such as PTBXL and WISDM. Further experiments with CNN, MLP, and RNN-based backbones are provided in the Appendix A.5.

### 4.2.2 VISUALIZATION WITH T-SNE.

Figure 2 presents a visualization of the feature representations learned using each SSDA method. The visualized features were extracted from the previous layer of each model classifier to ensure consistent comparison. For reference, the target-only model—trained with fully labeled target domain data—serves as an upper-bound representation. In cases where only 5% of the target domain

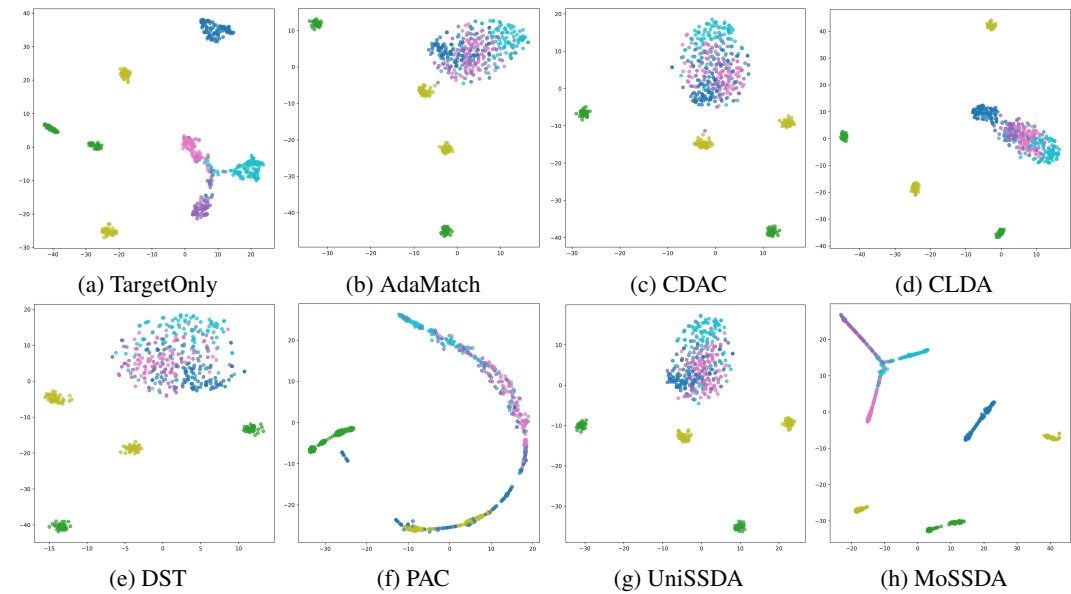

(a) TargetOnly     (b) AdaMatch     (c) CDAC     (d) CLDA

(e) DST     (f) PAC     (g) UniSSDA     (h) MoSSDA

Figure 2: t-SNE visualization learned on HHAR 1 to 6 DA scenario target test data, when using RESNET18 backbone and unlabeled ratio is 0.95

labels were available, our proposed method, MoSSDA method showed the closest alignment to the fully supervised target-only representation. Most benchmark methods failed to separate Class 0 features well, except for CLDA and MoSSDA. MoSSDA achieved clear separation across all six classes, whereas the other methods could not distinguish between classes 2, 3, and 5. These results suggest that MoSSDA effectively enhances the feature discriminability in the target domain under limited label supervision by leveraging labeled and unlabeled target data.

## 4.3 ABLATION STUDY

Table 3 reports ablation results on the MFD dataset (univariate, CNN backbone, unlabeled ratio 0.95, class imbalance 1:5:5), systematically evaluating the contribution of each MoSSDA component. The term "diff" indicates the performance gap between the full MoSSDA model and its ablated variants. Removing the positive supervised contrastive loss—with domain-cross mixup—caused the largest performance degradation, particularly impacting the F1-score. This highlights the critical role of discriminative feature learning in adapting to limited labeled target data.

The two-stage decoupled learning strategy demonstrated the second-largest effect, validating its design that separates representation learning from classifier training: the first stage learns class-discriminative features across intra- and inter-domain samples, and the second stage trains the classifier on the fixed feature space. Ablating mixup within the contrastive module showed a measurable performance drop, confirming that domain-cross mixup enhances positive contrastive learning by

| | $\mathcal{L}_{mmd}$ | $\mathcal{L}_{ctr}$ | mixup | 2-step | accuracy | | f1-score | |
|---|---|---|---|---|---|---|---|---|
| | | | | | Avg. | diff | Avg. | diff |
| Proposed | ✓ | ✓ | ✓ | ✓ | 0.9798 | - | 0.9736 | - |
| w/o mmd loss | | ✓ | ✓ | ✓ | 0.9710 | ↓0.009 | 0.9172 | ↓0.056 |
| w/o ctr loss | ✓ | | | ✓ | 0.4550 | ↓0.525 | 0.2085 | ↓0.765 |
| w/o phase1 mix | ✓ | ✓ | | ✓ | 0.8635 | ↓0.116 | 0.8472 | ↓0.126 |
| regular contrastive learning | ✓ | regular | ✓ | ✓ | 0.9260 | ↓0.054 | 0.8801 | ↓0.093 |
| w/o 2-stage learning | ✓ | ✓ | ✓ | | 0.8063 | ↓0.174 | 0.7805 | ↓0.193 |
| w/o momentum encoder | ✓ | ✓ | ✓ | ✓ | 0.9550 | ↓0.025 | 0.9011 | ↓0.073 |

Table 3: Ablation study on effectiveness of proposed methods, evaluated on MFD. The unlabeled ratio is 0.95 and backbone is CNN

enriching feature diversity. Similarly, excluding the MMD loss led to degradation due to weaker domain alignment in the backbone encoder. Additional experiments confirmed that incorporating a momentum encoder yields further gains. The proposed positive supervised contrastive loss with mixup consistently outperforms standard InfoNCE (Eq. 9)-based objectives, particularly under class imbalance favoring majority classes. Comparing to naive contrastive learning on all data, MoSSDA's tailored modules improve minority class recognition robustness and reduce imbalance-induced biases.

In several backbone based experiments (Table 3 and Appendix A.6), the positive contrastive module contributed the most to the overall adaptation performance, followed by the two-step decoupled learning framework. Overall, these results establish that the positive contrastive module with mixup across source and target domains contributes most significantly to overall adaptation performance, followed by the two-stage decoupled learning framework. Other components such as mixup and the domain-invariant encoder also provide meaningful benefits, especially in improving F1-score robustness under class imbalance.

## 5 CONCLUSION

In this study, we propose MoSSDA, a novel framework for the domain adaptation problem within the context of semi-supervised learning for multivariate time series classification. The MoSSDA features a simple yet effective decoupled learnable structure. Our approach combines mixup and positive supervised contrastive learning in feature space, allowing the model to distinguish between discriminative and consistent features despite highly limited annotations. This architecture incorporates a momentum encoder to ensure the stability and consistency of the learned feature representations, which is a critical factor for time-series data. The decoupled two-stage learning strategy improves the model robustness and generalization. In addition, our framework allows flexible integration of various backbone models. The proposed method outperformed state-of-the-art benchmark methods, including augmentation-based SSDA approaches, in highly unlabeled target domain scenarios. Extensive experiments on benchmark datasets validate its superiority.

MoSSDA achieves strong results on diverse time-series classification tasks but still faces several limitations. Minority-class performance can degrade under extreme long-tailed label distributions and skewed unlabeled target data, and the method remains sensitive to backbone architecture and training dynamics. Moreover, the framework assumes identical label spaces across domains and relies on fixed representations, limiting robustness under open-set or partial adaptation and evolving target distributions. Future work includes imbalance-aware and architecture-agnostic objectives, backbone-adaptive meta-learning, and few/zero-shot, unknown-class, and test-time adaptation mechanisms to improve robustness in realistic deployment scenarios.

**Reproducibility.** All of our implementation and code are available at `https://drive.google.com/drive/folders/1BqfQ3EEZME9_lizWVDXB5CBGuKoDpLXP`

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

## A APPENDIX

### A.1 DATASET DETAILS

**UCIHAR** Anguita et al. (2013) comprises sensor data collected from 30 subjects during six activities: walking, ascending stairs, descending stairs, standing, sitting, and lying down. The data acquisition process incorporated accelerometers, gyroscopes, and body sensors. Each subject is treated as a distinct domain to account for inter-subject variability.

**WISDM** Kwapisz et al. (2011) comprises accelerometer readings from 36 participants who performed the same six physical activities described for A.1. This dataset is notable for its ability to capture temporal variability, thereby facilitating the evaluation of domain adaptation methods in activity recognition tasks.

**HHAR** Stisen et al. (2015) encompasses the sensor signals from nine individuals' smartphones and smartwatches. The data for each subject constitutes a distinct domain, facilitating the exploration of heterogeneity in sensor-based human activity recognition.

**PTBXL** Wagner et al. (2022; 2020) is a substantial clinical dataset comprising 12-lead electrocardiogram (ECG) signals. The ECG recordings were obtained from 11 distinct ECG device models, of which the three most represented devices define unique domains. The classification task is organized into five diagnostic super-classes, with significant class imbalance and domain-distribution heterogeneity presenting notable challenges.

**EEG** Goldberger et al. (2000) under consideration includes single-channel recordings from 20 healthy subjects classified into five sleep stages: Wake, N1, N2, N3, and REM. Each individual's data constitutes a domain, thereby facilitating subject transfer scenarios in sleep stage classification.

**MFD** Lessmeier et al. (2016) comprises uni-variate vibration signals obtained under four distinct operating conditions, each treated as a distinct domain. The dataset is utilized to evaluate the efficacy of the model in the context of initial fault detection.

### A.2 BENCHMARK METHODS DETAILS

**AdaMatch** Berthelot et al. (2021) provides a unified approach for semi-supervised domain adaptation by applying both weak and strong augmentations to achieve effective distribution alignment between source and target data.

**CDAC** Li et al. (2021a) method addresses inter- and intra-domain adaptation by employing adversarial adaptive clustering loss and aligning feature clusters across domains. Pseudo-labeling is utilized to expand the set of labeled samples during training.

**DST** Chen et al. (2022) mitigates self-training bias in semi-supervised settings by decoupling pseudo-label generation and utilization across two classifier heads and adversarially optimizing feature representations to improve pseudo-label quality.

**PAC** Mishra et al. (2021) demonstrates that a robust target classifier can be obtained through self-supervised pretraining (e.g., rotation prediction) and consistency regularization, obviating the need for explicit source-target alignment in semi-supervised domain adaptation.

**UniSSDA** Zhang et al. (2024) addresses common-class bias in universal domain adaptation by introducing a prior-guided pseudo-label refinement strategy, supporting mixed private and common class scenarios for both source and target domains.

**CLDA** Singh (2021) is a single-stage contrastive learning framework comprising inter-domain contrastive alignment of class centroids and instance-level similarity maximization, thereby enhancing representation learning under semi-supervised domain adaptation

### A.3 IMPLEMENTATION DETAILS

#### A.3.1 MODEL BACKBONES

- **Convolutional Neural Network (CNN)**: The employed 1D-CNN architecture comprises three convolutional blocks, each integrating a convolutional layer, batch normalization, ReLU activation, and max pooling. The structure has been designed to extract sequential patterns from time series data.

- **RESNET18**: ResNet-18 for 1D data incorporates residual connections to facilitate deep network training by enabling information flow across layers. This design has been demonstrated to effectively mitigate vanishing gradient effects while concurrently enhancing the efficacy of feature learning in the context of time series analysis.

- **Temporal Convolutional Network (TCN)**: The TCN employs causal, dilated convolutions to capture long-range temporal dependencies in sequential data, while effectively preventing information leakage across temporal blocks.

#### A.3.2 AUGMENTATIONS

Since the data augmentations used in the benchmark methods are intended for images, we replaced them with augmentations appropriate for our multivariate time series implementation. We implemented a suite of augmentations tailored for time series data.:

- **TSRandomHorizontalFlip**: randomly reverses the temporal sequence 50 percent of the time.

- **RandomErasingTS**: zero-masks randomly selected segments to improve robustness to missing data.

- **RandAugmentTS**: applies random augmentations sequentially from a predefined pool. The number (n) and strength (m) of transformations are controlled.

- **AddNoise**: introduces Gaussian noise to the input sequence.

- **Scale**: modifies the signal amplitude.

- **TimeWarp**: applies nonlinear temporal warping based on a beta distribution.

- **Cutout1D**: masks the input sequence by setting values to zero, similar to RandomErasingTS.

- **Permute**: segments and shuffles ordered batches of the input sequence.

The augmentations are composed differently per phase. One transformation is used during training (n = 1, m = 9); two transformations are used during the strong augmentation phases (n = 2, m = 10); and no transformations are applied during validation or testing.

### A.4 FULL RESULTS OF BOUNDARY

In the context of domain adaptation, theoretical upper and lower bounds provide a rigorous characterization of the attainable performance when transferring knowledge from a source domain $S$ to a target domain $T$. The fundamental outcome of Ben-David et al. (2006) formally substantiates the relationship between the target error $\mathcal{E}_T(h)$ of a hypothesis $h$, the source error $\mathcal{E}_S(h)$, and the domain divergence:

$$\mathcal{E}_T(h) \leq \mathcal{E}_S(h) + \frac{1}{2}d_{\mathcal{H}\Delta\mathcal{H}}(S,T) + \lambda,$$

where $\mathcal{E}_S(h)$ is the source domain error, $d_{\mathcal{H}\Delta\mathcal{H}}(S,T)$ denotes the $\mathcal{H}\Delta\mathcal{H}$-divergence measuring the distributional discrepancy between domains with respect to hypothesis space $\mathcal{H}$, and $\lambda = \min_{h'\in\mathcal{H}}(\mathcal{E}_S(h') + \mathcal{E}_T(h'))$ represents the joint error of the ideal hypothesis on both domains.

We compare our method with three distinct conditions to evaluate its adaptability.

In this framework:

- The **Target Only** model, trained with full supervision on the target domain, defines the *theoretical upper bound*, capturing the minimum attainable target risk given complete target labels.

- The **No Adaptation** model, trained solely on labeled source data without adaptation, represents a *theoretical lower bound* on target performance, since no domain shift mitigation is applied.

- The **Labeled Only** model, which has been trained with full supervision on the target and source domains, has been demonstrated to capture all information from both domains.

The evaluation of empirical results (Table S1) in relation to these bounds facilitates a rigorous assessment of the effectiveness of the method's adaptation and its proximity to ideal target-domain test performance.

| | | Target Only | | | | No Adaptation | | | | Labeled Only | | | | OURS | | | |
|---|---|---|---|---|---|---|---|---|---|---|---|---|---|---|---|---|---|
| | | accuracy | | f1-score | | accuracy | | f1-score | | accuracy | | f1-score | | accuracy | | f1-score | |
| unlbl ratio | backbone | Avg. | std. | Avg. | std. | Avg. | std. | Avg. | std. | Avg. | std. | Avg. | std. | Avg. | std. | Avg. | std. |
| | | | | | | | | EEG | | | | | | | | | |
| 0.7 | CNN | **0.8620** | 0.04 | **0.7751** | ±0.05 | 0.6913 | ±0.12 | 0.6162 | ±0.11 | 0.8151 | ±0.06 | 0.7286 | ±0.04 | 0.8369 | ±0.02 | 0.7555 | ±0.04 |
| | RESNET18 | **0.8619** | 0.04 | **0.7750** | ±0.05 | 0.6812 | ±0.12 | 0.5731 | ±0.10 | 0.8105 | ±0.06 | 0.6942 | ±0.05 | 0.7910 | ±0.05 | 0.6862 | ±0.05 |
| | TCN | 0.4734 | 0.05 | 0.3501 | ±0.04 | 0.4475 | ±0.07 | 0.3353 | ±0.07 | **0.4919** | ±0.06 | **0.3784** | ±0.06 | 0.4863 | ±0.04 | 0.3739 | ±0.05 |
| 0.9 | CNN | **0.8620** | 0.04 | **0.7751** | ±0.05 | 0.6913 | ±0.12 | 0.6162 | ±0.11 | 0.7648 | ±0.08 | 0.6722 | ±0.07 | 0.8057 | ±0.04 | 0.7245 | ±0.05 |
| | RESNET18 | **0.8619** | 0.04 | **0.7750** | ±0.05 | 0.6812 | ±0.12 | 0.5731 | ±0.10 | 0.7558 | ±0.07 | 0.6354 | ±0.07 | 0.7553 | ±0.07 | 0.6552 | ±0.06 |
| | TCN | 0.4734 | 0.05 | 0.3501 | ±0.04 | 0.4475 | ±0.07 | 0.3353 | ±0.07 | 0.4711 | ±0.06 | 0.3592 | ±0.06 | 0.4803 | ±0.06 | 0.3596 | ±0.06 |
| 0.95 | CNN | **0.8620** | 0.04 | **0.7751** | ±0.05 | 0.6913 | ±0.12 | 0.6162 | ±0.11 | 0.6993 | ±0.11 | 0.6145 | ±0.11 | 0.7813 | ±0.05 | 0.6991 | ±0.05 |
| | RESNET18 | **0.8619** | 0.04 | **0.7750** | ±0.05 | 0.6812 | ±0.12 | 0.5731 | ±0.10 | 0.6787 | ±0.11 | 0.5594 | ±0.11 | 0.7328 | ±0.07 | 0.6244 | ±0.06 |
| | TCN | **0.4734** | 0.05 | 0.3501 | ±0.04 | 0.4475 | ±0.07 | 0.3353 | ±0.07 | 0.4454 | ±0.07 | 0.3346 | ±0.07 | 0.4695 | ±0.05 | 0.3597 | ±0.05 |
| | | | | | | | | HAR | | | | | | | | | |
| 0.7 | CNN | 0.9012 | ±0.27 | 0.8861 | ±0.31 | 0.8672 | ±0.12 | 0.8666 | ±0.12 | 0.8595 | ±0.14 | 0.8617 | ±0.13 | **0.9708** | ±0.03 | **0.9704** | ±0.03 |
| | RESNET18 | 0.8581 | ±0.25 | 0.8386 | ±0.30 | 0.8085 | ±0.15 | 0.7916 | ±0.17 | 0.8252 | ±0.15 | 0.8137 | ±0.16 | **0.9594** | ±0.04 | **0.9606** | ±0.04 |
| | TCN | 0.7920 | ±0.23 | 0.7707 | ±0.25 | 0.8532 | ±0.12 | 0.8507 | ±0.12 | 0.8614 | ±0.13 | 0.8566 | ±0.14 | **0.9444** | ±0.04 | **0.9395** | ±0.04 |
| 0.9 | CNN | 0.9012 | ±0.27 | 0.8861 | ±0.31 | 0.8672 | ±0.12 | 0.8666 | ±0.12 | 0.8595 | ±0.14 | 0.8617 | ±0.13 | **0.9647** | ±0.04 | **0.9645** | ±0.04 |
| | RESNET18 | 0.8581 | ±0.25 | 0.8386 | ±0.30 | 0.8085 | ±0.15 | 0.7916 | ±0.17 | 0.8252 | ±0.15 | 0.8137 | ±0.16 | **0.9376** | ±0.05 | **0.9392** | ±0.04 |
| | TCN | 0.7920 | ±0.23 | 0.7707 | ±0.25 | 0.8532 | ±0.12 | 0.8507 | ±0.12 | 0.8614 | ±0.13 | 0.8566 | ±0.14 | **0.8963** | ±0.08 | **0.8917** | ±0.08 |
| 0.95 | CNN | 0.9012 | ±0.27 | 0.8861 | ±0.31 | 0.8672 | ±0.12 | 0.8666 | ±0.12 | 0.8595 | ±0.14 | 0.8617 | ±0.13 | **0.9279** | ±0.09 | **0.9161** | ±0.11 |
| | RESNET18 | 0.8581 | ±0.25 | 0.8386 | ±0.30 | 0.8085 | ±0.15 | 0.7916 | ±0.17 | 0.8252 | ±0.15 | 0.8137 | ±0.16 | **0.8970** | ±0.10 | **0.8948** | ±0.10 |
| | TCN | 0.7920 | ±0.23 | 0.7707 | ±0.25 | 0.8532 | ±0.12 | 0.8507 | ±0.12 | 0.8614 | ±0.13 | **0.8566** | ±0.14 | 0.8604 | ±0.08 | 0.8516 | ±0.08 |
| | | | | | | | | HHAR | | | | | | | | | |
| 0.7 | CNN | **0.9868** | ±0.01 | **0.9869** | ±0.01 | 0.6830 | ±0.18 | 0.6542 | ±0.20 | 0.9788 | ±0.02 | 0.9791 | ±0.02 | 0.9784 | ±0.01 | 0.9788 | ±0.01 |
| | RESNET18 | **0.9653** | ±0.03 | 0.9665 | ±0.02 | 0.6815 | ±0.19 | 0.6569 | ±0.22 | 0.9529 | ±0.05 | 0.9455 | ±0.08 | **0.9693** | ±0.02 | **0.9698** | ±0.02 |
| | TCN | 0.9528 | ±0.03 | 0.9537 | ±0.03 | 0.6446 | ±0.20 | 0.6151 | ±0.22 | **0.9561** | ±0.02 | **0.9568** | ±0.02 | 0.9089 | ±0.06 | 0.9072 | ±0.07 |
| 0.9 | CNN | **0.9868** | ±0.01 | **0.9869** | ±0.01 | 0.6830 | ±0.18 | 0.6542 | ±0.20 | 0.9353 | ±0.04 | 0.9359 | ±0.04 | 0.9595 | ±0.02 | 0.9604 | ±0.02 |
| | RESNET18 | **0.9653** | ±0.03 | **0.9665** | ±0.02 | 0.6815 | ±0.19 | 0.6569 | ±0.22 | 0.9133 | ±0.04 | 0.9100 | ±0.05 | 0.9563 | ±0.02 | 0.9567 | ±0.02 |
| | TCN | 0.9528 | ±0.03 | 0.9537 | ±0.03 | 0.6446 | ±0.20 | 0.6151 | ±0.22 | 0.9076 | ±0.06 | 0.9073 | ±0.06 | 0.9013 | ±0.06 | 0.9020 | ±0.06 |
| 0.95 | CNN | **0.9868** | ±0.01 | **0.9869** | ±0.01 | 0.6830 | ±0.18 | 0.6542 | ±0.20 | 0.6995 | ±0.17 | 0.6670 | ±0.20 | 0.9480 | ±0.03 | 0.9494 | ±0.03 |
| | RESNET18 | **0.9653** | ±0.03 | **0.9665** | ±0.02 | 0.6815 | ±0.19 | 0.6569 | ±0.22 | 0.7094 | ±0.18 | 0.6819 | ±0.21 | 0.9430 | ±0.03 | 0.9433 | ±0.03 |
| | TCN | **0.9528** | ±0.03 | **0.9537** | ±0.03 | 0.6446 | ±0.20 | 0.6151 | ±0.22 | 0.6490 | ±0.20 | 0.6224 | ±0.22 | 0.8456 | ±0.13 | 0.8471 | ±0.13 |
| | | | | | | | | MFD | | | | | | | | | |
| 0.7 | CNN | 0.9639 | ±0.02 | 0.9734 | ±0.02 | 0.8175 | ±0.16 | 0.8082 | ±0.19 | 0.9692 | ±0.06 | 0.9732 | ±0.05 | **0.9832** | ±0.03 | **0.9793** | ±0.03 |
| | RESNET18 | **0.9995** | ±0.00 | **0.9996** | ±0.00 | 0.8339 | ±0.13 | 0.8271 | ±0.17 | 0.9848 | ±0.02 | 0.9863 | ±0.02 | 0.9726 | ±0.04 | 0.9571 | ±0.07 |
| | TCN | 0.5732 | ±0.04 | 0.5608 | ±0.05 | 0.5710 | ±0.04 | 0.5686 | ±0.06 | 0.5987 | ±0.03 | 0.6143 | ±0.04 | **0.6062** | ±0.03 | **0.6225** | ±0.04 |
| 0.9 | CNN | 0.9639 | ±0.02 | 0.9734 | ±0.02 | 0.8175 | ±0.16 | 0.8082 | ±0.19 | 0.9316 | ±0.10 | 0.9182 | ±0.14 | **0.9777** | ±0.05 | **0.9759** | ±0.05 |
| | RESNET18 | **0.9995** | ±0.00 | **0.9996** | ±0.00 | 0.8339 | ±0.13 | 0.8271 | ±0.17 | 0.9396 | ±0.10 | 0.9382 | ±0.10 | 0.9339 | ±0.11 | 0.9065 | ±0.15 |
| | TCN | 0.5732 | ±0.04 | 0.5608 | ±0.05 | 0.5710 | ±0.04 | 0.5686 | ±0.06 | 0.5891 | ±0.04 | 0.5938 | ±0.06 | **0.5923** | ±0.03 | **0.6062** | ±0.04 |
| 0.95 | CNN | 0.9639 | ±0.02 | 0.9734 | ±0.02 | 0.8175 | ±0.16 | 0.8082 | ±0.19 | 0.9191 | ±0.11 | 0.8971 | ±0.17 | **0.9798** | ±0.03 | **0.9736** | ±0.03 |
| | RESNET18 | **0.9995** | ±0.00 | **0.9996** | ±0.00 | 0.8339 | ±0.13 | 0.8271 | ±0.17 | 0.9263 | ±0.10 | 0.9193 | ±0.13 | 0.9519 | ±0.06 | 0.9096 | ±0.14 |
| | TCN | 0.5732 | ±0.04 | 0.5608 | ±0.05 | 0.5710 | ±0.04 | 0.5686 | ±0.06 | **0.5849** | ±0.05 | 0.5917 | ±0.08 | 0.5828 | ±0.03 | **0.6052** | ±0.04 |
| | | | | | | | | PTBXL | | | | | | | | | |
| 0.7 | CNN | **0.7555** | ±0.04 | **0.6179** | ±0.09 | 0.6773 | ±0.06 | 0.5343 | ±0.08 | 0.7246 | ±0.04 | 0.5736 | ±0.08 | 0.7294 | ±0.03 | 0.5962 | ±0.08 |
| | RESNET18 | **0.7802** | ±0.04 | **0.6639** | ±0.06 | 0.6978 | ±0.06 | 0.5848 | ±0.06 | 0.7568 | ±0.04 | 0.6358 | ±0.07 | 0.7361 | ±0.04 | 0.6179 | ±0.07 |
| | TCN | **0.4548** | ±0.04 | **0.2689** | ±0.05 | 0.4012 | ±0.05 | 0.2351 | ±0.02 | 0.4439 | ±0.02 | 0.2632 | ±0.05 | 0.4284 | ±0.03 | 0.2551 | ±0.04 |
| 0.9 | CNN | **0.7555** | ±0.04 | **0.6179** | ±0.09 | 0.6773 | ±0.06 | 0.5343 | ±0.08 | 0.7147 | ±0.03 | 0.5688 | ±0.07 | 0.7209 | ±0.03 | 0.5719 | ±0.07 |
| | RESNET18 | **0.7802** | ±0.04 | **0.6639** | ±0.06 | 0.6978 | ±0.06 | *0.5848* | ±0.06 | 0.7287 | ±0.05 | 0.6043 | ±0.07 | 0.7213 | ±0.02 | 0.5880 | ±0.04 |
| | TCN | **0.4548** | ±0.04 | **0.2689** | ±0.05 | 0.4012 | ±0.05 | 0.2351 | ±0.02 | 0.4108 | ±0.05 | 0.2389 | ±0.02 | 0.4389 | ±0.03 | 0.2459 | ±0.03 |
| 0.95 | CNN | **0.7555** | ±0.04 | **0.6179** | ±0.09 | 0.6773 | ±0.06 | 0.5343 | ±0.08 | 0.6840 | ±0.06 | 0.5510 | ±0.08 | 0.7005 | ±0.03 | 0.5632 | ±0.08 |
| | RESNET18 | **0.7802** | ±0.04 | **0.6639** | ±0.06 | 0.6978 | ±0.06 | 0.5848 | ±0.06 | 0.7118 | ±0.06 | 0.5983 | ±0.08 | 0.7014 | ±0.01 | 0.5701 | ±0.05 |
| | TCN | **0.4548** | ±0.04 | **0.2689** | ±0.05 | 0.4012 | ±0.05 | 0.2351 | ±0.02 | 0.4056 | ±0.05 | 0.2375 | ±0.02 | 0.4451 | ±0.04 | 0.2299 | ±0.03 |
| | | | | | | | | WISDM | | | | | | | | | |
| 0.7 | CNN | 0.9448 | ±0.04 | 0.8940 | ±0.10 | 0.6990 | ±0.16 | 0.5447 | ±0.17 | **0.9518** | 0.03 | **0.9106** | ±0.08 | 0.8357 | 0.15 | 0.7987 | ±0.18 |
| | RESNET18 | 0.8681 | ±0.07 | **0.7764** | ±0.12 | 0.5909 | ±0.15 | 0.4463 | ±0.16 | 0.8318 | 0.06 | 0.6972 | ±0.09 | 0.7838 | 0.04 | 0.7176 | ±0.08 |
| | TCN | 0.5482 | ±0.13 | 0.3193 | ±0.12 | 0.6649 | ±0.13 | 0.5037 | ±0.14 | **0.8472** | 0.04 | **0.7299** | ±0.09 | 0.8203 | 0.05 | 0.7020 | ±0.12 |
| 0.9 | CNN | 0.9448 | ±0.04 | 0.8940 | ±0.10 | 0.6990 | ±0.16 | 0.5447 | ±0.17 | 0.8944 | 0.03 | 0.7635 | ±0.11 | 0.7323 | 0.12 | 0.6536 | ±0.12 |
| | RESNET18 | 0.8681 | ±0.07 | **0.7764** | ±0.12 | 0.5909 | ±0.15 | 0.4463 | ±0.16 | 0.7458 | 0.08 | 0.6261 | ±0.14 | 0.7403 | 0.11 | 0.6709 | ±0.15 |
| | TCN | 0.5482 | ±0.13 | 0.3193 | ±0.12 | 0.6649 | ±0.13 | 0.5037 | ±0.14 | 0.7679 | 0.08 | 0.6415 | ±0.12 | 0.7811 | 0.06 | **0.6538** | ±0.10 |
| 0.95 | CNN | 0.9448 | ±0.04 | 0.8940 | ±0.10 | 0.6990 | ±0.16 | 0.5447 | ±0.17 | 0.8702 | 0.04 | 0.7189 | ±0.13 | 0.6459 | 0.17 | 0.5745 | ±0.18 |
| | RESNET18 | 0.8681 | ±0.07 | **0.7764** | ±0.12 | 0.5909 | ±0.15 | 0.4463 | ±0.16 | 0.7107 | 0.09 | 0.5895 | ±0.13 | 0.6732 | 0.13 | 0.6084 | ±0.16 |
| | TCN | 0.5482 | ±0.13 | 0.3193 | ±0.12 | 0.6649 | ±0.13 | 0.5037 | ±0.14 | **0.7490** | 0.07 | **0.6206** | ±0.13 | 0.7031 | 0.06 | 0.5943 | ±0.11 |

Table S1: Comparison with Boundary : Target domain test performances averaged across domain pairs for each datasets.

## A.5 FULL RESULTS OF BACKBONES

To further validate our approach, we extended our experiments by incorporating four additional backbones tailored for time-series data. These backbones include two recurrent neural network ar-

| unlbl ratio | metric | AdaMatch Avg. | std. | CDAC Avg. | std. | DST Avg. | std. | PAC Avg. | std. | UniSSDA Avg. | std. | CLDA Avg. | std. | OURS Avg. | std. |
|---|---|---|---|---|---|---|---|---|---|---|---|---|---|---|---|
| | | | | | | | | **EEG** | | | | | | | |
| 0.7 | acc | 0.4864 | ±0.06 | 0.2217 | ±0.16 | 0.4983 | ±0.08 | 0.6469 | ±0.11 | 0.4499 | ±0.03 | 0.3676 | ±0.14 | **0.8369** | ±0.02 |
| | f1 | 0.3816 | ±0.06 | 0.0768 | ±0.05 | 0.4017 | ±0.07 | 0.5353 | ±0.12 | 0.3382 | ±0.06 | 0.2500 | ±0.08 | **0.7555** | ±0.04 |
| 0.9 | acc | 0.4638 | ±0.06 | 0.2237 | ±0.16 | 0.4572 | ±0.08 | 0.5646 | ±0.09 | 0.4028 | ±0.06 | 0.3268 | ±0.14 | **0.8057** | ±0.04 |
| | f1 | 0.3755 | ±0.05 | 0.0779 | ±0.06 | 0.3865 | ±0.07 | 0.4338 | ±0.10 | 0.3147 | ±0.07 | 0.2296 | ±0.07 | **0.7245** | ±0.05 |
| 0.95 | acc | 0.4576 | ±0.06 | 0.2219 | ±0.16 | 0.4522 | ±0.08 | 0.4957 | ±0.14 | 0.3974 | ±0.06 | 0.3111 | ±0.13 | **0.7813** | ±0.05 |
| | f1 | 0.3718 | ±0.05 | 0.0770 | ±0.06 | 0.3879 | ±0.07 | 0.3501 | ±0.11 | 0.3121 | ±0.07 | 0.2115 | ±0.06 | **0.6991** | ±0.05 |
| | | | | | | | | **HAR** | | | | | | | |
| 0.7 | acc | 0.5020 | ±0.10 | 0.1513 | ±0.06 | 0.5142 | ±0.13 | 0.6838 | ±0.21 | 0.4372 | ±0.07 | 0.4525 | ±0.12 | **0.9708** | ±0.03 |
| | f1 | 0.3873 | ±0.09 | 0.0430 | ±0.02 | 0.4071 | ±0.11 | 0.6273 | ±0.25 | 0.3238 | ±0.07 | 0.3227 | ±0.10 | **0.9704** | ±0.03 |
| 0.9 | acc | 0.5071 | ±0.11 | 0.1513 | ±0.06 | 0.5008 | ±0.10 | 0.5076 | ±0.23 | 0.4240 | ±0.06 | 0.4539 | ±0.11 | **0.9647** | ±0.04 |
| | f1 | 0.3936 | ±0.11 | 0.0430 | ±0.02 | 0.3999 | ±0.09 | 0.4113 | ±0.27 | 0.3096 | ±0.06 | 0.3140 | ±0.09 | **0.9645** | ±0.04 |
| 0.95 | acc | 0.4947 | ±0.11 | 0.1513 | ±0.06 | 0.4993 | ±0.10 | 0.5567 | ±0.16 | 0.4155 | ±0.07 | 0.4611 | ±0.13 | **0.9279** | ±0.09 |
| | f1 | 0.3696 | ±0.09 | 0.0430 | ±0.02 | 0.3958 | ±0.08 | 0.4666 | ±0.20 | 0.3022 | ±0.05 | 0.3225 | ±0.11 | **0.9161** | ±0.11 |
| | | | | | | | | **HHAR** | | | | | | | |
| 0.7 | acc | 0.4380 | ±0.15 | 0.1383 | ±0.04 | 0.4393 | ±0.15 | 0.5897 | ±0.23 | 0.4407 | ±0.16 | 0.4195 | ±0.19 | **0.9784** | ±0.01 |
| | f1 | 0.3897 | ±0.17 | 0.0414 | ±0.01 | 0.3986 | ±0.18 | 0.5504 | ±0.27 | 0.3945 | ±0.17 | 0.3813 | ±0.17 | **0.9788** | ±0.01 |
| 0.9 | acc | 0.4416 | ±0.16 | 0.1412 | ±0.03 | 0.4525 | ±0.17 | 0.5330 | ±0.25 | 0.4541 | ±0.17 | 0.4370 | ±0.22 | **0.9595** | ±0.02 |
| | f1 | 0.3973 | ±0.18 | 0.0423 | ±0.01 | 0.4131 | ±0.18 | 0.4740 | ±0.27 | 0.4118 | ±0.19 | 0.4022 | ±0.20 | **0.9604** | ±0.02 |
| 0.95 | acc | 0.4415 | ±0.16 | 0.1421 | ±0.03 | 0.4499 | ±0.16 | 0.4197 | ±0.18 | 0.4457 | ±0.17 | 0.4325 | ±0.22 | **0.9480** | ±0.03 |
| | f1 | 0.3957 | ±0.18 | 0.0425 | ±0.01 | 0.4113 | ±0.18 | 0.3492 | ±0.19 | 0.4011 | ±0.18 | 0.4013 | ±0.20 | **0.9494** | ±0.03 |
| | | | | | | | | **MFD** | | | | | | | |
| 0.7 | acc | 0.5544 | ±0.12 | 0.4539 | ±0.00 | 0.5775 | ±0.10 | 0.7957 | ±0.18 | 0.5707 | ±0.10 | 0.5577 | ±0.04 | **0.9832** | ±0.03 |
| | f1 | 0.6036 | ±0.17 | 0.2081 | ±0.00 | 0.6183 | ±0.19 | 0.7843 | ±0.25 | 0.6153 | ±0.17 | 0.4977 | ±0.11 | **0.9793** | ±0.03 |
| 0.9 | acc | 0.5501 | ±0.12 | 0.4539 | ±0.00 | 0.6953 | ±0.27 | 0.6882 | ±0.10 | 0.5538 | ±0.10 | 0.5679 | ±0.04 | **0.9777** | ±0.05 |
| | f1 | 0.6008 | ±0.17 | 0.2081 | ±0.00 | 0.6824 | ±0.31 | 0.6784 | ±0.11 | 0.6037 | ±0.18 | 0.5047 | ±0.05 | **0.9759** | ±0.05 |
| 0.95 | acc | 0.5547 | ±0.12 | 0.4539 | ±0.00 | 0.5676 | ±0.04 | 0.7331 | ±0.15 | 0.5596 | ±0.10 | 0.5721 | ±0.04 | **0.9798** | ±0.03 |
| | f1 | 0.6048 | ±0.18 | 0.2081 | ±0.00 | 0.5904 | ±0.06 | 0.7529 | ±0.16 | 0.6079 | ±0.18 | 0.5090 | ±0.05 | **0.9736** | ±0.03 |
| | | | | | | | | **PTBXL** | | | | | | | |
| 0.7 | acc | 0.4520 | ±0.15 | 0.1732 | ±0.10 | 0.5082 | ±0.06 | 0.5841 | ±0.12 | 0.5018 | ±0.06 | 0.5690 | ±0.05 | **0.7294** | ±0.03 |
| | f1 | 0.1910 | ±0.02 | 0.0632 | ±0.04 | 0.2132 | ±0.03 | 0.4432 | ±0.05 | 0.1916 | ±0.04 | 0.2748 | ±0.05 | **0.5962** | ±0.08 |
| 0.9 | acc | 0.4454 | ±0.17 | 0.1735 | ±0.10 | 0.5061 | ±0.06 | 0.5944 | ±0.05 | 0.4914 | ±0.07 | 0.4868 | ±0.15 | **0.7209** | ±0.03 |
| | f1 | 0.2043 | ±0.03 | 0.0642 | ±0.04 | 0.2276 | ±0.03 | 0.4040 | ±0.07 | 0.1885 | ±0.04 | 0.2325 | ±0.04 | **0.5719** | ±0.07 |
| 0.95 | acc | 0.4459 | ±0.17 | 0.1737 | ±0.10 | 0.5025 | ±0.06 | 0.4857 | ±0.07 | 0.4061 | ±0.16 | 0.4754 | ±0.16 | **0.7005** | ±0.03 |
| | f1 | 0.2166 | ±0.04 | 0.0650 | ±0.04 | 0.2303 | ±0.03 | 0.2917 | ±0.02 | 0.1715 | ±0.06 | 0.2209 | ±0.04 | **0.5632** | ±0.08 |
| | | | | | | | | **WISDM** | | | | | | | |
| 0.7 | acc | 0.3422 | ±0.10 | 0.1344 | ±0.16 | 0.3306 | ±0.16 | 0.5591 | ±0.14 | 0.3348 | ±0.14 | 0.2444 | ±0.24 | **0.8357** | ±0.15 |
| | f1 | 0.0923 | ±0.03 | 0.0349 | ±0.04 | 0.2195 | ±0.13 | 0.3202 | ±0.15 | 0.2221 | ±0.11 | 0.1577 | ±0.18 | **0.7987** | ±0.18 |
| 0.9 | acc | 0.3020 | ±0.12 | 0.1344 | ±0.16 | 0.3365 | ±0.09 | 0.5964 | ±0.16 | 0.3274 | ±0.12 | 0.2262 | ±0.22 | **0.7323** | ±0.12 |
| | f1 | 0.0795 | ±0.02 | 0.0349 | ±0.04 | 0.2399 | ±0.10 | 0.3886 | ±0.18 | 0.2172 | ±0.11 | 0.1476 | ±0.17 | **0.6536** | ±0.12 |
| 0.95 | acc | 0.2795 | ±0.19 | 0.1344 | ±0.16 | 0.3074 | ±0.12 | 0.4632 | ±0.20 | 0.3075 | ±0.12 | 0.2070 | ±0.17 | **0.6459** | ±0.17 |
| | f1 | 0.0988 | ±0.08 | 0.0349 | ±0.04 | 0.2585 | ±0.14 | 0.3017 | ±0.22 | 0.2145 | ±0.10 | 0.1532 | ±0.13 | **0.5745** | ±0.18 |

Table S2: **Comparison with SSDA methods**: Averaged target domain test accuracy and f1-score across domain pairs for each datasets with CNN backbone. The best performance is in bold and the second best is underlined.

chitectures—GRU and LSTM—and two multi-layer perceptron-based models—NLinear and DLinear. The Figure S1 illustrates the performance on the HAR dataset, evaluated on the target domain test set under three different unlabeled data ratios: 0.7, 0.9, and 0.95. This comparison highlights how each backbone performs under increasing scarcity of labeled target data. With the TCN Table S3, a prevalent approach for time-series data, MoSSDA combines the optimal overall performance, followed by CLDA, which utilizes contrastive loss. These findings emphasize the necessity of selecting an appropriate backbone for the dataset to attain optimal domain adaptation performance.

A.6 FULL ABLATION STUDY RESULTS

Tables S4, S5, S6, and S7 present the results of ablation studies across four different time-series datasets: HHAR for human activity recognition, PTBXL for electrocardiogram (ECG) based diagnosis, EEG for sleep stage prediction, and MFD for machining fault classification. In each case, experiments were performed under two fixed unlabeled ratios (0.9 and 0.95), and compared across two backbone architectures—CNN and ResNet18. These results provide insight into the contribution of each component in our method and the influence of different backbone choices across various datasets.

In the regular contrastive learning baseline, InfoNCE loss is applied jointly across all features from source and target domains, as follows:

$$\mathcal{L}_{\text{InfoNCE}} = -\frac{1}{N} \sum_{i=1}^{N} \log \frac{\sum_{j=1}^{N} \mathbf{1}_{y_i=y_j} \exp\left(\frac{\text{sim}(\mathbf{z}_i, \mathbf{z}_j)}{\tau}\right)}{\sum_{j=1}^{N} \exp\left(\frac{\text{sim}(\mathbf{z}_i, \mathbf{z}_j)}{\tau}\right)} \quad (9)$$

| unlbl ratio | metric | AdaMatch Avg. | std. | CDAC Avg. | std. | DST Avg. | std. | PAC Avg. | std. | UniSSDA Avg. | std. | CLDA Avg. | std. | OURS Avg. | std. |
|---|---|---|---|---|---|---|---|---|---|---|---|---|---|---|---|
| | | | | | | **EEG** | | | | | | | | | |
| 0.7 | acc | 0.3465 | ±0.05 | 0.1295 | ±0.09 | 0.3509 | ±0.06 | 0.3489 | ±0.11 | 0.2575 | ±0.07 | 0.2973 | ±0.10 | **0.4863** | ±0.04 |
| | f1 | 0.2558 | ±0.04 | 0.0652 | ±0.04 | 0.2536 | ±0.04 | 0.1965 | ±0.13 | 0.1660 | ±0.04 | 0.1946 | ±0.05 | **0.3739** | ±0.05 |
| 0.9 | acc | 0.3089 | ±0.05 | 0.1297 | ±0.09 | 0.3146 | ±0.05 | 0.3084 | ±0.09 | 0.2468 | ±0.07 | 0.2700 | ±0.11 | **0.4803** | ±0.06 |
| | f1 | 0.2524 | ±0.04 | 0.0652 | ±0.04 | 0.2499 | ±0.03 | 0.1718 | ±0.07 | 0.1589 | ±0.04 | 0.1900 | ±0.06 | **0.3596** | ±0.06 |
| 0.95 | acc | 0.3001 | ±0.05 | 0.1291 | ±0.09 | 0.3081 | ±0.05 | 0.3137 | ±0.11 | 0.2423 | ±0.07 | 0.2586 | ±0.10 | **0.4695** | ±0.05 |
| | f1 | 0.2469 | ±0.03 | 0.0649 | ±0.04 | 0.2483 | ±0.03 | 0.1525 | ±0.08 | 0.1565 | ±0.04 | 0.1883 | ±0.06 | **0.3597** | ±0.05 |
| | | | | | | **HAR** | | | | | | | | | |
| 0.7 | acc | 0.6240 | ±0.09 | 0.1476 | ±0.03 | 0.6287 | ±0.11 | 0.1846 | ±0.03 | 0.6227 | ±0.10 | 0.6002 | ±0.15 | **0.9444** | ±0.04 |
| | f1 | 0.5498 | ±0.08 | 0.0572 | ±0.03 | 0.5458 | ±0.13 | 0.0623 | ±0.04 | 0.5295 | ±0.11 | 0.5489 | ±0.15 | **0.9395** | ±0.04 |
| 0.9 | acc | 0.6487 | ±0.07 | 0.1487 | ±0.03 | 0.6324 | ±0.11 | 0.1727 | ±0.03 | 0.6072 | ±0.09 | 0.6036 | ±0.15 | **0.8963** | ±0.08 |
| | f1 | 0.5677 | ±0.08 | 0.0574 | ±0.03 | 0.5568 | ±0.11 | 0.0490 | ±0.01 | 0.5259 | ±0.11 | 0.5501 | ±0.17 | **0.8917** | ±0.08 |
| 0.95 | acc | 0.6400 | ±0.07 | 0.1488 | ±0.03 | 0.6358 | ±0.10 | 0.1683 | ±0.02 | 0.6037 | ±0.08 | 0.5867 | ±0.15 | **0.8604** | ±0.08 |
| | f1 | 0.5478 | ±0.08 | 0.0574 | ±0.03 | 0.5605 | ±0.12 | 0.0479 | ±0.01 | 0.5151 | ±0.11 | 0.5277 | ±0.16 | **0.8516** | ±0.08 |
| | | | | | | **HHAR** | | | | | | | | | |
| 0.7 | acc | 0.5598 | ±0.18 | 0.1542 | ±0.04 | 0.5728 | ±0.18 | 0.2363 | ±0.14 | 0.5740 | ±0.19 | 0.6137 | ±0.19 | **0.9089** | ±0.06 |
| | f1 | 0.5384 | ±0.21 | 0.0656 | ±0.03 | 0.5487 | ±0.22 | 0.1133 | ±0.18 | 0.5486 | ±0.22 | 0.5747 | ±0.21 | **0.9072** | ±0.07 |
| 0.9 | acc | 0.5574 | ±0.18 | 0.1524 | ±0.04 | 0.5697 | ±0.19 | 0.2903 | ±0.15 | 0.5668 | ±0.19 | 0.6042 | ±0.20 | **0.9013** | ±0.06 |
| | f1 | 0.5379 | ±0.21 | 0.0644 | ±0.03 | 0.5494 | ±0.23 | 0.1926 | ±0.17 | 0.5442 | ±0.22 | 0.5688 | ±0.22 | **0.9020** | ±0.06 |
| 0.95 | acc | 0.5529 | ±0.18 | 0.1534 | ±0.04 | 0.5688 | ±0.19 | 0.3078 | ±0.14 | 0.5709 | ±0.19 | 0.6061 | ±0.21 | **0.8456** | ±0.13 |
| | f1 | 0.5346 | ±0.21 | 0.0655 | ±0.03 | 0.5495 | ±0.22 | 0.1990 | ±0.18 | 0.5487 | ±0.22 | 0.5703 | ±0.22 | **0.8471** | ±0.13 |
| | | | | | | **MFD** | | | | | | | | | |
| 0.7 | acc | 0.5617 | ±0.03 | 0.4550 | ±0.00 | 0.5744 | ±0.10 | 0.0910 | ±0.00 | 0.5625 | ±0.04 | 0.5404 | ±0.06 | **0.6062** | ±0.03 |
| | f1 | 0.5725 | ±0.04 | 0.2085 | ±0.00 | 0.6168 | ±0.19 | 0.0556 | ±0.00 | 0.5720 | ±0.04 | 0.5556 | ±0.07 | **0.6225** | ±0.04 |
| 0.9 | acc | 0.5599 | ±0.03 | 0.4550 | ±0.00 | **0.6970** | ±0.28 | 0.0910 | ±0.00 | 0.5433 | ±0.04 | 0.5149 | ±0.06 | 0.5923 | ±0.03 |
| | f1 | 0.5733 | ±0.04 | 0.2085 | ±0.00 | **0.6824** | ±0.32 | 0.0556 | ±0.00 | 0.5621 | ±0.05 | 0.5177 | ±0.07 | 0.6062 | ±0.04 |
| 0.95 | acc | 0.5575 | ±0.03 | 0.4550 | ±0.00 | 0.5630 | ±0.05 | 0.0910 | ±0.00 | 0.5436 | ±0.04 | 0.5103 | ±0.05 | **0.5828** | ±0.03 |
| | f1 | 0.5711 | ±0.04 | 0.2085 | ±0.00 | 0.5861 | ±0.06 | 0.0556 | ±0.00 | 0.5618 | ±0.05 | 0.5106 | ±0.07 | **0.6052** | ±0.04 |
| | | | | | | **PTBXL** | | | | | | | | | |
| 0.7 | acc | 0.4519 | ±0.09 | 0.1683 | ±0.03 | 0.4169 | ±0.11 | 0.3264 | ±0.19 | **0.4522** | ±0.12 | 0.4088 | ±0.08 | 0.4284 | ±0.03 |
| | f1 | 0.1981 | ±0.02 | 0.1177 | ±0.04 | 0.1853 | ±0.01 | 0.1209 | ±0.05 | 0.1657 | ±0.01 | 0.2346 | ±0.01 | **0.2551** | ±0.04 |
| 0.9 | acc | 0.3998 | ±0.09 | 0.1680 | ±0.03 | 0.3264 | ±0.06 | 0.2316 | ±0.18 | 0.4236 | ±0.17 | 0.3277 | ±0.05 | **0.4389** | ±0.03 |
| | f1 | 0.2144 | ±0.01 | 0.1187 | ±0.04 | 0.2004 | ±0.02 | 0.0701 | ±0.04 | 0.1603 | ±0.02 | 0.2329 | ±0.02 | **0.2459** | ±0.03 |
| 0.95 | acc | 0.3670 | ±0.09 | 0.1683 | ±0.03 | 0.2971 | ±0.05 | **0.4788** | ±0.08 | 0.4191 | ±0.17 | 0.2861 | ±0.04 | 0.4451 | ±0.04 |
| | f1 | 0.2203 | ±0.02 | 0.1183 | ±0.04 | 0.2032 | ±0.03 | 0.1519 | ±0.03 | 0.1638 | ±0.03 | 0.2241 | ±0.03 | **0.2299** | ±0.03 |
| | | | | | | **WISDM** | | | | | | | | | |
| 0.7 | acc | 0.3314 | ±0.08 | 0.1132 | ±0.10 | 0.3133 | ±0.09 | 0.1808 | ±0.12 | 0.2130 | ±0.18 | 0.3586 | ±0.08 | **0.8203** | ±0.05 |
| | f1 | 0.1057 | ±0.04 | 0.0435 | ±0.04 | 0.1057 | ±0.03 | 0.0485 | ±0.03 | 0.0535 | ±0.04 | 0.3050 | ±0.10 | **0.7020** | ±0.12 |
| 0.9 | acc | 0.3055 | ±0.13 | 0.1184 | ±0.10 | 0.2945 | ±0.11 | 0.1274 | ±0.09 | 0.1385 | ±0.12 | 0.3430 | ±0.07 | **0.7811** | ±0.06 |
| | f1 | 0.1032 | ±0.07 | 0.0471 | ±0.04 | 0.1190 | ±0.08 | 0.0392 | ±0.03 | 0.0379 | ±0.03 | 0.2950 | ±0.10 | **0.6538** | ±0.10 |
| 0.95 | acc | 0.2950 | ±0.13 | 0.1184 | ±0.09 | 0.2796 | ±0.12 | 0.1320 | ±0.10 | 0.1059 | ±0.10 | 0.3252 | ±0.08 | **0.7031** | ±0.06 |
| | f1 | 0.1212 | ±0.08 | 0.0467 | ±0.04 | 0.1230 | ±0.08 | 0.0371 | ±0.02 | 0.0302 | ±0.02 | 0.2725 | ±0.09 | **0.5943** | ±0.11 |

Table S3: **Comparison with SSDA methods**: Averaged target domain test accuracy and f1-score across domain pairs for each datasets with TCN backbone. The best performance is in bold and the second best is underlined.

| | mmd_loss | ctr_loss | phase1 mix | 2-step | backbone | unlbl_ratio = 0.9 accuracy Avg. | std. | f1-score Avg. | std. | unlbl_ratio = 0.95 accuracy Avg. | std. | f1-score Avg. | std. |
|---|---|---|---|---|---|---|---|---|---|---|---|---|---|
| Proposed | ✓ | ✓ | ✓ | ✓ | CNN | 0.9595 | ±0.02 | 0.9604 | ±0.02 | 0.9480 | ±0.03 | 0.9494 | ±0.03 |
| | | | | | RESNET18 | 0.9563 | ±0.02 | 0.9567 | ±0.02 | 0.9430 | ±0.03 | 0.9433 | ±0.03 |
| w/o mmd loss | | ✓ | ✓ | ✓ | CNN | 0.9584 | ±0.03 | 0.9594 | ±0.03 | 0.9443 | ±0.03 | 0.9451 | ±0.03 |
| | | | | | RESNET18 | 0.9479 | ±0.03 | 0.9492 | ±0.02 | 0.9340 | ±0.03 | 0.9344 | ±0.03 |
| w/o ctr loss | ✓ | | | ✓ | CNN | 0.2028 | ±0.01 | 0.0562 | ±0.00 | 0.2039 | ±0.01 | 0.0564 | ±0.00 |
| | | | | | RESNET18 | 0.2039 | ±0.01 | 0.0564 | ±0.00 | 0.2018 | ±0.01 | 0.0559 | ±0.00 |
| w/o phase1 mix | ✓ | ✓ | | ✓ | CNN | 0.8885 | ±0.11 | 0.8819 | ±0.13 | 0.8597 | ±0.11 | 0.8538 | ±0.12 |
| | | | | | RESNET18 | 0.9463 | ±0.03 | 0.9465 | ±0.03 | 0.8875 | ±0.10 | 0.8813 | ±0.10 |
| regular contrastive learning | ✓ | regular | | ✓ | CNN | 0.9074 | ±0.14 | 0.8665 | ±0.22 | 0.9260 | ±0.11 | 0.8801 | ±0.20 |
| | | | | | RESNET18 | 0.9186 | ±0.12 | 0.8855 | ±0.20 | 0.8939 | ±0.15 | 0.8616 | ±0.22 |
| w/o 2-stage learning | ✓ | ✓ | ✓ | | CNN | 0.5713 | ±0.12 | 0.5429 | ±0.14 | 0.4805 | ±0.11 | 0.4479 | ±0.12 |
| | | | | | RESNET18 | 0.4371 | ±0.08 | 0.4212 | ±0.09 | 0.3642 | ±0.06 | 0.3412 | ±0.06 |
| w/o momentum encoder | ✓ | ✓ | ✓ | ✓ | CNN | 0.9562 | ±0.06 | 0.9000 | ±0.15 | 0.9550 | ±0.07 | 0.9011 | ±0.16 |
| | | | | | RESNET18 | 0.9525 | ±0.06 | 0.8940 | ±0.17 | 0.9508 | ±0.06 | 0.8839 | ±0.17 |

Table S4: Ablation study on effectiveness of proposed methods, evaluated on HHAR dataset across domain pairs.

where $\mathbf{z}_i$ is the normalized feature vector, $y_i$ is the class label, $\tau$ is the temperature parameter, $N$ is the total number of samples from source and target batches, and $\mathrm{sim}(\cdot, \cdot)$ denotes dot-product similarity. Here, we set $\tau = 0.07$.

| | mmd_loss | ctr_loss | phase1 mix | 2-step | backbone | unlbl_ratio = 0.9 | | | | unlbl_ratio = 0.95 | | | |
|---|---|---|---|---|---|---|---|---|---|---|---|---|---|
| | | | | | | accuracy | | f1-score | | accuracy | | f1-score | |
| | | | | | | Avg. | std. | Avg. | std. | Avg. | std. | Avg. | std. |
| Proposed | ✓ | ✓ | ✓ | ✓ | CNN | 0.7209 | ±0.03 | 0.5719 | ±0.07 | 0.7005 | ±0.03 | 0.5632 | ±0.08 |
| | | | | | RESNET18 | 0.7213 | ±0.02 | 0.5880 | ±0.04 | 0.7014 | ±0.01 | 0.5701 | ±0.05 |
| w/o mmd loss | | ✓ | ✓ | ✓ | CNN | 0.7029 | ±0.01 | 0.5490 | ±0.06 | 0.7088 | ±0.03 | 0.5525 | ±0.07 |
| | | | | | RESNET18 | 0.7184 | ±0.02 | 0.5825 | ±0.06 | 0.6976 | ±0.02 | 0.5645 | ±0.05 |
| w/o ctr loss | ✓ | | | ✓ | CNN | 0.5147 | ±0.09 | 0.1352 | ±0.02 | 0.5147 | ±0.09 | 0.1352 | ±0.02 |
| | | | | | RESNET18 | 0.5147 | ±0.09 | 0.1352 | ±0.02 | 0.5147 | ±0.09 | 0.1352 | ±0.02 |
| w/o phase1 mix | ✓ | ✓ | | ✓ | CNN | 0.7070 | ±0.05 | 0.5584 | ±0.07 | 0.6921 | ±0.05 | 0.5475 | ±0.08 |
| | | | | | RESNET18 | 0.7032 | ±0.02 | 0.5585 | ±0.07 | 0.6828 | ±0.02 | 0.5273 | ±0.06 |
| regular contrastive learning | ✓ | regular | | ✓ | CNN | 0.7083 | ±0.05 | 0.5470 | ±0.08 | 0.6978 | ±0.05 | 0.5403 | ±0.09 |
| | | | | | RESNET18 | 0.6949 | ±0.03 | 0.5539 | ±0.07 | 0.6896 | ±0.02 | 0.5466 | ±0.05 |
| w/o 2-stage learning | ✓ | ✓ | ✓ | | CNN | 0.4802 | ±0.13 | 0.3354 | ±0.09 | 0.5341 | ±0.09 | 0.3788 | ±0.07 |
| | | | | | RESNET18 | 0.5988 | ±0.04 | 0.4351 | ±0.03 | 0.5876 | ±0.05 | 0.4195 | ±0.04 |
| w/o momentum encoder | ✓ | ✓ | ✓ | ✓ | CNN | 0.7086 | ±0.04 | 0.5597 | ±0.08 | 0.6903 | ±0.03 | 0.5452 | ±0.07 |
| | | | | | RESNET18 | 0.7051 | ±0.03 | 0.5713 | ±0.05 | 0.6900 | ±0.03 | 0.5535 | ±0.06 |

Table S5: Ablation study on effectiveness of proposed methods, evaluated on PTBXL dataset across domain pairs.

| | mmd_loss | ctr_loss | phase1 mix | 2-step | backbone | unlbl_ratio = 0.9 | | | | unlbl_ratio = 0.95 | | | |
|---|---|---|---|---|---|---|---|---|---|---|---|---|---|
| | | | | | | accuracy | | f1-score | | accuracy | | f1-score | |
| | | | | | | Avg. | std. | Avg. | std. | Avg. | std. | Avg. | std. |
| Proposed | ✓ | ✓ | ✓ | ✓ | CNN | 0.8057 | ±0.04 | 0.7245 | ±0.05 | 0.7813 | ±0.05 | 0.6991 | ±0.05 |
| | | | | | RESNET18 | 0.7553 | ±0.07 | 0.6552 | ±0.06 | 0.7328 | ±0.07 | 0.6244 | ±0.06 |
| w/o mmd loss | | ✓ | ✓ | ✓ | CNN | 0.7820 | ±0.06 | 0.6921 | ±0.07 | 0.7721 | ±0.07 | 0.5525 | ±0.07 |
| | | | | | RESNET18 | 0.7734 | ±0.07 | 0.6439 | ±0.07 | 0.7437 | ±0.07 | 0.5645 | ±0.05 |
| w/o ctr loss | ✓ | | | ✓ | CNN | 0.4160 | ±0.06 | 0.1171 | ±0.01 | 0.4160 | ±0.06 | 0.1352 | ±0.02 |
| | | | | | RESNET18 | 0.4160 | ±0.06 | 0.1171 | ±0.01 | 0.4160 | ±0.06 | 0.1352 | ±0.02 |
| w/o phase1 mix | ✓ | ✓ | | ✓ | CNN | 0.7679 | ±0.04 | 0.6472 | ±0.04 | 0.7392 | ±0.03 | 0.5475 | ±0.08 |
| | | | | | RESNET18 | 0.7398 | ±0.07 | 0.5970 | ±0.08 | 0.7123 | ±0.09 | 0.5273 | ±0.06 |
| regular contrastive learning | ✓ | regular | | ✓ | CNN | 0.7632 | ±0.05 | 0.6648 | ±0.05 | 0.7527 | ±0.04 | 0.6587 | ±0.05 |
| | | | | | RESNET18 | 0.7568 | ±0.06 | 0.6400 | ±0.04 | 0.7340 | ±0.07 | 0.6175 | ±0.06 |
| w/o 2-stage learning | ✓ | ✓ | ✓ | | CNN | 0.6232 | ±0.08 | 0.4596 | ±0.10 | 0.5557 | ±0.11 | 0.3788 | ±0.07 |
| | | | | | RESNET18 | 0.5851 | ±0.06 | 0.4510 | ±0.07 | 0.5538 | ±0.07 | 0.4195 | ±0.04 |
| w/o momentum encoder | ✓ | ✓ | ✓ | ✓ | CNN | 0.7847 | ±0.05 | 0.7018 | ±0.05 | 0.7799 | ±0.04 | 0.6950 | ±0.05 |
| | | | | | RESNET18 | 0.7628 | ±0.05 | 0.6763 | ±0.06 | 0.7344 | ±0.06 | 0.6424 | ±0.06 |

Table S6: Ablation study on effectiveness of proposed methods, evaluated on EEG dataset across domain pairs.

| | mmd_loss | ctr_loss | phase1 mix | 2-step | backbone | unlbl_ratio = 0.9 | | | | unlbl_ratio = 0.95 | | | |
|---|---|---|---|---|---|---|---|---|---|---|---|---|---|
| | | | | | | accuracy | | f1-score | | accuracy | | f1-score | |
| | | | | | | Avg. | std. | Avg. | std. | Avg. | std. | Avg. | std. |
| Proposed | ✓ | ✓ | ✓ | ✓ | CNN | 0.9777 | ±0.05 | 0.9759 | ±0.05 | 0.9798 | ±0.03 | 0.9736 | ±0.03 |
| | | | | | RESNET18 | 0.9339 | ±0.11 | 0.9065 | ±0.15 | 0.9519 | ±0.06 | 0.9096 | ±0.14 |
| w/o mmd loss | | ✓ | ✓ | ✓ | CNN | 0.9651 | ±0.05 | 0.9246 | ±0.13 | 0.9710 | ±0.04 | 0.9172 | ±0.14 |
| | | | | | RESNET18 | 0.9581 | ±0.06 | 0.9306 | ±0.10 | 0.9448 | ±0.06 | 0.8661 | ±0.16 |
| w/o ctr loss | ✓ | | | ✓ | CNN | 0.4539 | ±0.00 | 0.2081 | ±0.00 | 0.4550 | ±0.00 | 0.2085 | ±0.00 |
| | | | | | RESNET18 | 0.4539 | ±0.00 | 0.2081 | ±0.00 | 0.4539 | ±0.00 | 0.2081 | ±0.00 |
| w/o phase1 mix | ✓ | ✓ | | ✓ | CNN | 0.8842 | ±0.08 | 0.8449 | ±0.15 | 0.8635 | ±0.11 | 0.8472 | ±0.15 |
| | | | | | RESNET18 | 0.9171 | ±0.08 | 0.8998 | ±0.12 | 0.8912 | ±0.09 | 0.8992 | ±0.08 |
| regular contrastive learning | ✓ | regular | | ✓ | CNN | 0.9074 | ±0.14 | 0.8665 | ±0.22 | 0.9260 | ±0.11 | 0.8801 | ±0.20 |
| | | | | | RESNET18 | 0.9186 | ±0.12 | 0.8855 | ±0.20 | 0.8939 | ±0.15 | 0.8616 | ±0.22 |
| w/o 2-stage learning | ✓ | ✓ | ✓ | | CNN | 0.7862 | ±0.17 | 0.7604 | ±0.21 | 0.8063 | ±0.13 | 0.7805 | ±0.16 |
| | | | | | RESNET18 | 0.7483 | ±0.15 | 0.6853 | ±0.18 | 0.7219 | ±0.19 | 0.6542 | ±0.20 |
| w/o momentum encoder | ✓ | ✓ | ✓ | ✓ | CNN | 0.9562 | ±0.06 | 0.9000 | ±0.15 | 0.9550 | ±0.07 | 0.9011 | ±0.16 |
| | | | | | RESNET18 | 0.9525 | ±0.06 | 0.8940 | ±0.17 | 0.9508 | ±0.06 | 0.8839 | ±0.17 |

Table S7: Ablation study on effectiveness of proposed methods, evaluated on MFD dataset across domain pairs.

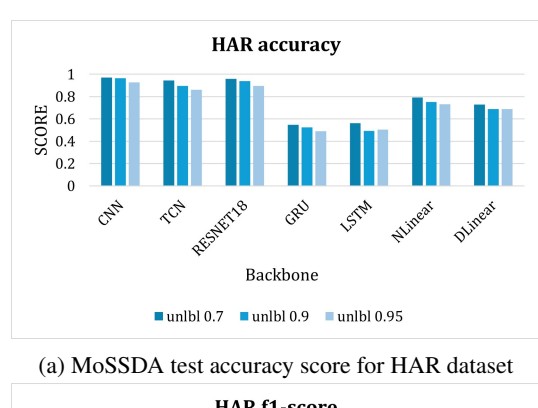

(a) MoSSDA test accuracy score for HAR dataset

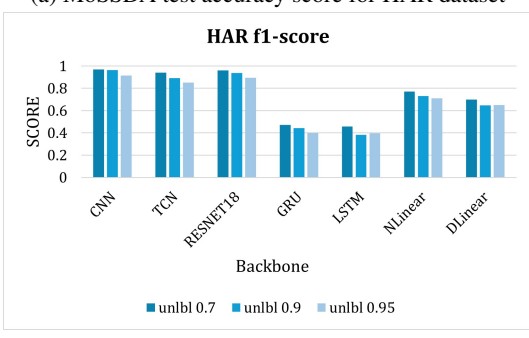

(b) MoSSDA test f1-score score for HAR dataset

Figure S1: Comparison with different backbone networks.

### A.7 FULL RESULTS OF VARIOUS EXPERIMENTAL SETTINGS

#### A.7.1 RESULTS OF HYPERPARAMETER COMBINATIONS

Tables S8 and S9 summarize the results of an extensive greedy search over the hyperparamters for weighting the multi-objective loss components (Eq.7) in MoSSDA, specifically the MMD loss ($\lambda_{mmd}$) and positive contrastive loss ($\lambda_{ctr}$). For each backbone (CNN and RESNET18) and representative datasets (EEG, HHAR, MFD), we varied $\lambda_{mmd}$ and $\lambda_{ctr}$ in $\{0.3, 0.5, 0.7\}$ across several unlabeled ratios.

Overall, the results demonstrate the MoSSDA's performance is marginally affected by reasonable variations in loss weighting within the tested range. The best scores in each dataset and unlabeled ratio do not differ substantially from those in equally balanced settings ($\lambda_{mmd} = 0.5, \lambda_{ctr} = 0.5$), which we adopt as the primary configuration throughout the main experiments for robust and fair comparisons. The minor differences observed across hyperparameter settings provide empirical evidence for MoSSDA's stability under multi-objective optimization, supportin gpractical deployment without requring extensive tuning.

Table S10 presents a comprehensive comparison of MoSSDA's performance across varying momentum coefficient ($m \in \{0.9, 0.99, 0.999\}$) for the momentum encoder, evaluated on multiple backbones and datasets with different unlabeled ratios. The momentum encoder stabilizes feature representations used for keys in contrastive learning, mitigating rapid shifts that could otherwise destabilize learning and reduce final accuracy. However, excessively high momentum values may oversmooth domain-specific features, particularly in the target domain, thereby risking loss of valuable fine-grained information critical for adaptation.

As shown in Table S10, models with a momentum coefficient of 0.99 consistently achieved the best or near-best accuracy and F1-score across most backbone and dataset configurations, particularly under high class imbalance and limited labeled data. These results indicate that an $m$ value of 0.99 effectively balances feature consistency preservation with adequate adaptation to target-domain nuances. Based on these findings, we adopt $m = 0.99$ as the default configuration for all main experi-

| | | mmd_w=0.3 | | | | mmd_w=0.5 | | | | mmd_w=0.7 | | | |
|---|---|---|---|---|---|---|---|---|---|---|---|---|---|
| | | accuracy | | f1-score | | accuracy | | f1-score | | accuracy | | f1-score | |
| unlbl | ctr_w | Avg. | std. | Avg. | std. | Avg. | std. | Avg. | std. | Avg. | std. | Avg. | std. |
| | | EEG | | | | | | | | | | | |
| | 0.3 | 0.7979 | 0.05 | 0.7156 | ±0.06 | 0.7975 | ±0.05 | 0.7158 | ±0.05 | 0.7955 | ±0.05 | 0.7161 | ±0.05 |
| 0.9 | 0.5 | 0.8021 | 0.05 | **0.7205** | ±0.05 | 0.7978 | ±0.05 | 0.7154 | ±0.06 | 0.7962 | ±0.05 | 0.7173 | ±0.05 |
| | 0.7 | **0.8034** | 0.05 | 0.7201 | ±0.05 | 0.8014 | ±0.05 | 0.7188 | ±0.06 | 0.7989 | ±0.05 | 0.7182 | ±0.06 |
| | 0.3 | 0.7800 | 0.05 | 0.6917 | ±0.06 | 0.7696 | ±0.06 | 0.6834 | ±0.06 | 0.7695 | ±0.06 | 0.6777 | ±0.07 |
| 0.95 | 0.5 | 0.7783 | 0.05 | 0.6877 | ±0.06 | 0.7806 | ±0.05 | 0.6909 | ±0.06 | 0.7695 | ±0.04 | 0.6830 | ±0.05 |
| | 0.7 | **0.7886** | 0.05 | **0.6987** | ±0.06 | 0.7802 | ±0.05 | 0.6925 | ±0.06 | 0.7806 | ±0.05 | 0.6922 | ±0.06 |
| | | HHAR | | | | | | | | | | | |
| | 0.3 | 0.9683 | 0.02 | 0.9688 | ±0.02 | 0.9672 | ±0.02 | 0.9677 | ±0.02 | 0.9684 | ±0.02 | 0.9690 | ±0.02 |
| 0.9 | 0.5 | 0.9688 | 0.02 | 0.9694 | ±0.02 | 0.9687 | ±0.02 | 0.9693 | ±0.02 | 0.9666 | ±0.02 | 0.9671 | ±0.02 |
| | 0.7 | **0.9701** | 0.02 | **0.9705** | ±0.02 | 0.9692 | ±0.02 | 0.9697 | ±0.02 | 0.9690 | ±0.02 | 0.9695 | ±0.02 |
| | 0.3 | 0.9597 | 0.02 | 0.9607 | ±0.02 | 0.9598 | ±0.02 | 0.9607 | ±0.02 | **0.9618** | ±0.02 | **0.9626** | ±0.02 |
| 0.95 | 0.5 | 0.9541 | 0.02 | 0.9550 | ±0.02 | 0.9584 | ±0.02 | 0.9593 | ±0.02 | 0.9585 | ±0.02 | 0.9595 | ±0.02 |
| | 0.7 | 0.9581 | 0.02 | 0.9592 | ±0.02 | 0.9577 | ±0.02 | 0.9587 | ±0.02 | 0.9583 | ±0.02 | 0.9593 | ±0.02 |
| | | MFD | | | | | | | | | | | |
| | 0.3 | 0.9266 | 0.11 | 0.9038 | ±0.17 | 0.9221 | ±0.12 | 0.8986 | ±0.17 | 0.9193 | ±0.12 | 0.8762 | ±0.21 |
| 0.9 | 0.5 | **0.9286** | 0.10 | 0.9077 | ±0.16 | 0.9186 | ±0.13 | 0.8798 | ±0.20 | 0.9161 | ±0.13 | 0.8798 | ±0.20 |
| | 0.7 | 0.9237 | 0.11 | **0.9079** | ±0.15 | 0.9214 | ±0.12 | 0.8795 | ±0.20 | 0.9171 | ±0.13 | 0.8739 | ±0.21 |
| | 0.3 | 0.8881 | 0.17 | 0.8504 | ±0.24 | 0.8796 | ±0.17 | 0.8443 | ±0.25 | **0.8975** | ±0.15 | 0.8587 | ±0.23 |
| 0.95 | 0.5 | 0.8909 | 0.16 | 0.8585 | ±0.23 | 0.8879 | ±0.17 | 0.8508 | ±0.25 | 0.8888 | ±0.16 | 0.8501 | ±0.24 |
| | 0.7 | 0.8928 | 0.16 | 0.8513 | ±0.23 | **0.9023** | ±0.14 | **0.8671** | ±0.22 | 0.8895 | ±0.17 | 0.8520 | ±0.24 |

Table S8: MoSSDA performances of different hyperparameter settings in CNN backbone. Bold represents best score in the same dataset and unlabeled ratio.

| | | mmd_w=0.3 | | | | mmd_w=0.5 | | | | mmd_w=0.7 | | | |
|---|---|---|---|---|---|---|---|---|---|---|---|---|---|
| | | accuracy | | f1-score | | accuracy | | f1-score | | accuracy | | f1-score | |
| unlbl | ctr_w | Avg. | std. | Avg. | std. | Avg. | std. | Avg. | std. | Avg. | std. | Avg. | std. |
| | | EEG | | | | | | | | | | | |
| | 0.3 | 0.7012 | 0.07 | 0.5898 | ±0.06 | 0.7175 | ±0.08 | 0.6120 | ±0.08 | 0.7182 | ±0.08 | 0.6135 | ±0.07 |
| 0.9 | 0.5 | 0.6938 | 0.08 | 0.5882 | ±0.08 | 0.7012 | ±0.09 | 0.5950 | ±0.08 | **0.7278** | ±0.08 | **0.6195** | ±0.07 |
| | 0.7 | 0.6880 | 0.08 | 0.5713 | ±0.06 | 0.7259 | ±0.05 | 0.6126 | ±0.06 | 0.7027 | ±0.07 | 0.5952 | ±0.07 |
| | 0.3 | 0.7079 | 0.08 | 0.5931 | ±0.08 | 0.7183 | ±0.08 | 0.6044 | ±0.07 | 0.7106 | ±0.07 | 0.5981 | ±0.08 |
| 0.95 | 0.5 | 0.7184 | 0.06 | 0.5970 | ±0.06 | **0.7234** | ±0.06 | **0.6100** | ±0.06 | 0.7112 | ±0.09 | 0.5969 | ±0.09 |
| | 0.7 | 0.6990 | 0.06 | 0.5736 | ±0.05 | 0.7221 | ±0.07 | 0.6037 | ±0.07 | 0.7135 | ±0.07 | 0.5987 | ±0.06 |
| | | HHAR | | | | | | | | | | | |
| | 0.3 | 0.9491 | 0.02 | 0.9498 | ±0.02 | 0.9506 | ±0.02 | 0.9512 | ±0.02 | 0.9525 | ±0.02 | 0.9531 | ±0.02 |
| 0.9 | 0.5 | 0.9506 | 0.02 | 0.9515 | ±0.02 | 0.9544 | ±0.02 | 0.9552 | ±0.02 | 0.9544 | ±0.02 | **0.9554** | ±0.02 |
| | 0.7 | 0.9456 | 0.03 | 0.9459 | ±0.03 | 0.9531 | ±0.02 | 0.9536 | ±0.02 | **0.9546** | ±0.02 | 0.9553 | ±0.02 |
| | 0.3 | 0.9394 | 0.04 | 0.9406 | ±0.04 | 0.9485 | ±0.02 | 0.9495 | ±0.02 | 0.9407 | ±0.02 | 0.9422 | ±0.02 |
| 0.95 | 0.5 | 0.9375 | 0.04 | 0.9387 | ±0.04 | 0.9453 | ±0.03 | 0.9464 | ±0.03 | **0.9494** | ±0.02 | **0.9505** | ±0.02 |
| | 0.7 | 0.9468 | 0.02 | 0.9478 | ±0.02 | 0.9463 | ±0.03 | 0.9473 | ±0.03 | 0.9448 | ±0.03 | 0.9460 | ±0.03 |
| | | MFD | | | | | | | | | | | |
| | 0.3 | 0.8895 | 0.11 | 0.8271 | ±0.19 | 0.8943 | ±0.10 | 0.8238 | ±0.19 | 0.8799 | ±0.11 | 0.8086 | ±0.20 |
| 0.9 | 0.5 | **0.9033** | 0.10 | **0.8492** | ±0.19 | 0.8814 | ±0.11 | 0.7986 | ±0.21 | 0.8906 | ±0.10 | 0.8273 | ±0.18 |
| | 0.7 | 0.8880 | 0.11 | 0.8285 | ±0.19 | 0.8970 | ±0.10 | 0.8254 | ±0.20 | 0.8857 | ±0.11 | 0.8151 | ±0.20 |
| | 0.3 | 0.9031 | 0.11 | 0.8578 | ±0.20 | 0.8980 | ±0.11 | 0.8299 | ±0.20 | 0.8978 | ±0.09 | 0.8389 | ±0.18 |
| 0.95 | 0.5 | 0.8985 | 0.12 | **0.8605** | ±0.21 | **0.9083** | ±0.10 | 0.8526 | ±0.20 | 0.9010 | ±0.11 | 0.8376 | ±0.21 |
| | 0.7 | 0.8959 | 0.11 | 0.8323 | ±0.20 | 0.8991 | ±0.12 | 0.8508 | ±0.21 | 0.8825 | ±0.12 | 0.7983 | ±0.22 |

Table S9: MoSSDA performances of different hyperparameter settings in RESNET18 backbone. Bold represents best score in the same dataset and unlabeled ratio.

| | | unlbl = 0.9 | | | | unlbl = 0.95 | | | |
| | | accuracy | | f1-score | | accuracy | | f1-score | |
| m | backbone | Avg. | std. | Avg. | std. | Avg. | std. | Avg. | std. |
|---|---|---|---|---|---|---|---|---|---|
| | | **EEG** | | | | | | | |
| 0.9 | CNN | 0.7972 | 0.05 | 0.7152 | ±0.06 | 0.7800 | 0.06 | 0.6897 | ±0.06 |
| | RESNET18 | **0.7118** | 0.08 | **0.6056** | ±0.07 | 0.7180 | 0.07 | 0.6065 | ±0.07 |
| 0.99 | CNN | 0.7978 | 0.05 | **0.7154** | ±0.06 | **0.7806** | 0.05 | **0.6909** | ±0.06 |
| | RESNET18 | 0.7012 | 0.09 | 0.5950 | ±0.08 | **0.7234** | 0.06 | **0.6100** | ±0.06 |
| 0.999 | CNN | **0.7979** | 0.05 | 0.7148 | ±0.06 | 0.7775 | 0.05 | 0.6863 | ±0.06 |
| | RESNET18 | 0.7075 | 0.09 | 0.6031 | ±0.08 | 0.7212 | 0.08 | 0.5971 | ±0.08 |
| | | **HAR** | | | | | | | |
| 0.9 | CNN | **0.9614** | ±0.05 | **0.9629** | ±0.05 | 0.9593 | ±0.06 | 0.9567 | ±0.06 |
| | RESNET18 | 0.9380 | ±0.05 | 0.9394 | ±0.04 | 0.9280 | ±0.07 | 0.9284 | ±0.07 |
| 0.99 | CNN | 0.9591 | ±0.05 | 0.9609 | ±0.04 | 0.9583 | ±0.06 | 0.9558 | ±0.06 |
| | RESNET18 | **0.9573** | ±0.04 | **0.9585** | ±0.04 | **0.9341** | ±0.06 | **0.9320** | ±0.07 |
| 0.999 | CNN | 0.9530 | ±0.05 | 0.9553 | ±0.05 | **0.9624** | ±0.05 | **0.9609** | ±0.05 |
| | RESNET18 | 0.9391 | ±0.06 | 0.9413 | ±0.05 | 0.9186 | ±0.12 | 0.9151 | ±0.13 |
| | | **HHAR** | | | | | | | |
| 0.9 | CNN | **0.9694** | ±0.02 | **0.9700** | ±0.02 | **0.9594** | ±0.02 | **0.9604** | ±0.02 |
| | RESNET18 | 0.9501 | ±0.03 | 0.9507 | ±0.03 | 0.9364 | ±0.04 | 0.9378 | ±0.04 |
| 0.99 | CNN | 0.9687 | ±0.02 | 0.9693 | ±0.02 | 0.9584 | ±0.02 | 0.9593 | ±0.02 |
| | RESNET18 | 0.9544 | ±0.02 | 0.9552 | ±0.02 | 0.9453 | ±0.03 | 0.9464 | ±0.03 |
| 0.999 | CNN | 0.9664 | ±0.02 | 0.9669 | ±0.02 | 0.9588 | ±0.02 | 0.9598 | ±0.02 |
| | RESNET18 | **0.9546** | ±0.02 | **0.9555** | ±0.02 | **0.9470** | ±0.03 | **0.9481** | ±0.03 |
| | | **MFD** | | | | | | | |
| 0.9 | CNN | **0.9235** | ±0.12 | **0.8837** | ±0.20 | 0.8846 | ±0.17 | 0.8473 | ±0.25 |
| | RESNET18 | 0.8902 | ±0.11 | 0.8199 | ±0.20 | 0.8989 | ±0.11 | 0.8522 | ±0.19 |
| 0.99 | CNN | 0.9186 | ±0.13 | 0.8798 | ±0.20 | **0.8879** | ±0.17 | **0.8508** | ±0.25 |
| | RESNET18 | 0.8814 | ±0.11 | 0.7986 | ±0.21 | **0.9083** | ±0.10 | 0.8526 | ±0.20 |
| 0.999 | CNN | 0.9173 | ±0.13 | 0.8784 | ±0.21 | 0.8831 | ±0.18 | 0.8475 | ±0.25 |
| | RESNET18 | **0.8928** | ±0.10 | **0.8228** | ±0.20 | 0.9000 | ±0.11 | **0.8611** | ±0.20 |
| | | **PTBXL** | | | | | | | |
| 0.9 | CNN | **0.7131** | ±0.01 | **0.5748** | ±0.05 | 0.6946 | ±0.02 | 0.5538 | ±0.07 |
| | RESNET18 | 0.6814 | ±0.04 | 0.5472 | ±0.07 | **0.6766** | ±0.03 | 0.5456 | ±0.06 |
| 0.99 | CNN | 0.7107 | ±0.02 | 0.5720 | ±0.06 | **0.6976** | ±0.01 | **0.5577** | ±0.07 |
| | RESNET18 | 0.6875 | ±0.03 | **0.5628** | ±0.05 | 0.6764 | ±0.03 | **0.5496** | ±0.06 |
| 0.999 | CNN | 0.7093 | ±0.02 | 0.5708 | ±0.06 | 0.6933 | ±0.02 | 0.5522 | ±0.08 |
| | RESNET18 | **0.6926** | ±0.04 | 0.5588 | ±0.06 | 0.6759 | ±0.02 | 0.5465 | ±0.05 |
| | | **WISDM** | | | | | | | |
| 0.9 | CNN | 0.8756 | ±0.06 | **0.7934** | ±0.13 | 0.8543 | ±0.08 | 0.7841 | ±0.13 |
| | RESNET18 | 0.7134 | ±0.10 | 0.6411 | ±0.15 | 0.6792 | ±0.08 | **0.6297** | ±0.12 |
| 0.99 | CNN | **0.8761** | ±0.08 | 0.7911 | ±0.15 | **0.8602** | ±0.06 | 0.7748 | ±0.15 |
| | RESNET18 | **0.7225** | ±0.13 | 0.6554 | ±0.17 | **0.6921** | ±0.07 | 0.6159 | ±0.12 |
| 0.999 | CNN | 0.8731 | ±0.08 | 0.7925 | ±0.14 | 0.8596 | ±0.07 | **0.7894** | ±0.13 |
| | RESNET18 | 0.7152 | ±0.13 | **0.6639** | ±0.15 | 0.6757 | ±0.06 | 0.6050 | ±0.11 |

Table S10: MoSSDA performances of different momentum for momentum encoder settings. Bold represents best score in the same condition.

ments. This choice ensures robust, stable training dynamics, supporting reliable domain adaptation in semi-supervised scenarios without extensive hyperparameter tuning.

### A.7.2 KERNEL SENSITIVITY ANALYSIS

Table S11 and S12 report MoSSDA's performance using various kernel functions for the MMD loss term, spanning two RBF kernels ($\sigma \in \{1, 0.5\}$) and two polynomial kernels (rank = 2, 3) against the default linear kernel. Results cover several datasets and both CNN and RESNET18 backbones at different unlabeled ratios.

Our experiments demonstrate that the linear kernel consistently yields the highest accuracy and f1-scores across most datasets and conditions. These results likely reflect the powerful nonlinear transformations already achieved by the domain-invariant encoder; by the time feature representations reach the MMD term, they are often well linearly separable. Accordingly, using a linear kernel preserves computational efficiency and keeps GPU usage tractable, a notable advantage in SSDA settings wehre labeled data are scarce.

| unlbl ratio | metric | Linear Avg. | std. | poly (rank=2) Avg. | std. | poly (rank=3) Avg. | std. | rbf ($\sigma$=1) Avg. | std. | rbf ($\sigma$=0.5) Avg. | std. |
|---|---|---|---|---|---|---|---|---|---|---|---|
| | | | | | EEG | | | | | | |
| 0.9 | accuracy | **0.8057** | ±0.04 | 0.8013 | ±0.05 | 0.7859 | ±0.07 | 0.7730 | ±0.05 | 0.7867 | ±0.05 |
| | f1-score | **0.7245** | ±0.05 | 0.7183 | ±0.05 | 0.6918 | ±0.06 | 0.7026 | ±0.05 | 0.7134 | ±0.05 |
| 0.95 | accuracy | 0.7813 | ±0.05 | **0.7845** | ±0.04 | 0.7722 | ±0.06 | 0.7350 | ±0.04 | 0.7657 | ±0.04 |
| | f1-score | **0.6991** | ±0.05 | 0.7006 | ±0.04 | 0.6812 | ±0.06 | 0.6727 | ±0.05 | 0.6872 | ±0.05 |
| | | | | | HAR | | | | | | |
| 0.9 | accuracy | **0.9647** | ±0.04 | 0.9473 | ±0.08 | 0.7653 | ±0.12 | 0.9479 | ±0.07 | 0.9468 | ±0.07 |
| | f1-score | **0.9645** | ±0.04 | 0.9454 | ±0.08 | 0.7123 | ±0.16 | 0.9457 | ±0.08 | 0.9475 | ±0.07 |
| 0.95 | accuracy | 0.9279 | ±0.09 | 0.9357 | ±0.09 | 0.7607 | ±0.17 | 0.9430 | ±0.07 | **0.9465** | ±0.07 |
| | f1-score | 0.9161 | ±0.11 | 0.9341 | ±0.09 | 0.6900 | ±0.22 | 0.9413 | ±0.08 | **0.9448** | ±0.08 |
| | | | | | HHAR | | | | | | |
| 0.9 | accuracy | 0.9595 | ±0.02 | **0.9759** | ±0.01 | 0.9700 | ±0.02 | 0.9692 | ±0.01 | 0.9702 | ±0.02 |
| | f1-score | 0.9604 | ±0.02 | **0.9763** | ±0.01 | 0.9705 | ±0.02 | 0.9699 | ±0.01 | 0.9706 | ±0.02 |
| 0.95 | accuracy | 0.9480 | ±0.03 | **0.9717** | ±0.02 | 0.9657 | ±0.02 | 0.9150 | ±0.11 | 0.9682 | ±0.02 |
| | f1-score | 0.9494 | ±0.03 | **0.9722** | ±0.02 | 0.9662 | ±0.02 | 0.9107 | ±0.12 | 0.9689 | ±0.02 |
| | | | | | MFD | | | | | | |
| 0.9 | accuracy | **0.9777** | ±0.05 | 0.9341 | ±0.09 | 0.9314 | ±0.10 | 0.8593 | ±0.11 | 0.9237 | ±0.09 |
| | f1-score | **0.9759** | ±0.05 | 0.9029 | ±0.15 | 0.8923 | ±0.16 | 0.8097 | ±0.14 | 0.8945 | ±0.15 |
| 0.95 | accuracy | **0.9798** | ±0.03 | 0.9179 | ±0.12 | 0.8970 | ±0.13 | 0.8662 | ±0.07 | 0.8719 | ±0.12 |
| | f1-score | **0.9736** | ±0.03 | 0.8756 | ±0.20 | 0.8295 | ±0.22 | 0.8207 | ±0.09 | 0.8165 | ±0.20 |
| | | | | | PTBXL | | | | | | |
| 0.9 | accuracy | **0.7209** | ±0.03 | 0.6935 | ±0.03 | 0.6735 | ±0.05 | 0.6945 | ±0.02 | 0.6890 | ±0.04 |
| | f1-score | 0.5719 | ±0.07 | 0.5804 | ±0.06 | 0.5349 | ±0.04 | **0.5846** | ±0.05 | 0.5670 | ±0.03 |
| 0.95 | accuracy | **0.7005** | ±0.03 | 0.6891 | ±0.03 | 0.6701 | ±0.04 | 0.6872 | ±0.02 | 0.6878 | ±0.02 |
| | f1-score | 0.5632 | ±0.08 | **0.5679** | ±0.07 | 0.5458 | ±0.08 | 0.5601 | ±0.03 | 0.5591 | ±0.03 |
| | | | | | WISDM | | | | | | |
| 0.9 | accuracy | 0.7323 | ±0.12 | **0.8257** | ±0.08 | 0.6785 | ±0.12 | 0.6855 | ±0.13 | 0.7511 | ±0.12 |
| | f1-score | 0.6536 | ±0.12 | **0.7057** | ±0.15 | 0.4905 | ±0.16 | 0.5418 | ±0.15 | 0.6953 | ±0.12 |
| 0.95 | accuracy | 0.6459 | ±0.17 | **0.7552** | ±0.11 | 0.6929 | ±0.15 | 0.5866 | ±0.16 | 0.7442 | ±0.11 |
| | f1-score | 0.5745 | ±0.18 | 0.6733 | ±0.15 | 0.5113 | ±0.23 | 0.5351 | ±0.09 | **0.6846** | ±0.10 |

Table S11: MoSSDA performances of different kernel for mmd loss in CNN backbone. Bold represents best score in the same condition.

While the linear kernel is generally robust and competitive, kernel choice can still be sensitive to specific dataset characteristics. For cases where the linear choice is sufficient, we encourage practitioners to consider RBF or custom kernels tailored to the structure of the dataset, balancing adaptation quality with resource constraints.

## A.8 PERFORMANCE DETAILS ON FULL SCENARIOS

Table S13 summarizes the complete cross-domain performance of MoSSDA on the PTBXL dataset. Since PTBXL defines three domains, all six possible source-target pairings were evaluated, allowing for a comprehensive view of generalizability across domain shifts. Similarly, Table S14 reports MoSSDA's performance on 10 selected source-target domain pairs out of the 36 defined combinations in the WISDM dataset. These results collectively assess the scalability and domain transferability of our proposed method under realistic and diverse deployment conditions.

## A.9 AUGMENTATION ABLATION FOR BASELINES

We compare several augmentation variants used in AdaMatch, UniSSDA, and DST, including input-level mixup.

In Tables S15 and S16, the "w/ mixup" rows (highlighted in gray) report results where the weak augmentation in the original baseline is replaced with input-level mixup, while the strong augmentation is kept unchanged. This variant is implemented in the standard way, mixing both inputs and labels as in conventional mixup.

When using a CNN backbone, input-level mixup improves performance on the HAR and HHAR datasets, and AdaMatch achieves its best results when mixup is included as part of the augmentation strategy. In contrast, with a ResNet-18 backbone, the original setting without input-level mixup

| unlbl ratio | metric | Linear Avg. | std. | poly (rank=2) Avg. | std. | poly (rank=3) Avg. | std. | rbf ($\sigma$=1) Avg. | std. | rbf ($\sigma$=0.5) Avg. | std. |
|---|---|---|---|---|---|---|---|---|---|---|---|
| | | | | | EEG | | | | | | |
| 0.9 | accuracy | 0.7553 | ±0.07 | **0.7622** | ±0.06 | 0.7588 | ±0.05 | 0.7348 | ±0.06 | 0.7614 | ±0.05 |
| | f1-score | 0.6552 | ±0.06 | 0.6741 | ±0.05 | **0.6856** | ±0.05 | 0.6726 | ±0.06 | 0.6593 | ±0.04 |
| 0.95 | accuracy | 0.7328 | ±0.07 | 0.7485 | ±0.05 | 0.7497 | ±0.05 | 0.7383 | ±0.05 | **0.7499** | ±0.05 |
| | f1-score | 0.6244 | ±0.06 | 0.6599 | ±0.05 | 0.6660 | ±0.05 | **0.6706** | ±0.05 | 0.6466 | ±0.04 |
| | | | | | HAR | | | | | | |
| 0.9 | accuracy | **0.9376** | ±0.05 | 0.9200 | ±0.11 | 0.8147 | ±0.14 | 0.9243 | ±0.06 | 0.9345 | ±0.06 |
| | f1-score | **0.9392** | ±0.04 | 0.9114 | ±0.13 | 0.7986 | ±0.16 | 0.9148 | ±0.08 | 0.9363 | ±0.06 |
| 0.95 | accuracy | 0.8970 | ±0.10 | 0.8896 | ±0.11 | 0.7618 | ±0.16 | **0.9301** | ±0.06 | 0.9002 | ±0.09 |
| | f1-score | 0.8948 | ±0.10 | 0.8759 | ±0.13 | 0.7349 | ±0.18 | **0.9256** | ±0.07 | 0.8964 | ±0.10 |
| | | | | | HHAR | | | | | | |
| 0.9 | accuracy | **0.9563** | ±0.02 | 0.9283 | ±0.04 | 0.9228 | ±0.05 | 0.9344 | ±0.02 | 0.9057 | ±0.02 |
| | f1-score | **0.9567** | ±0.02 | 0.9295 | ±0.04 | 0.9238 | ±0.05 | 0.9351 | ±0.02 | 0.9070 | ±0.02 |
| 0.95 | accuracy | **0.9430** | ±0.03 | 0.9122 | ±0.05 | 0.8964 | ±0.05 | 0.8991 | ±0.10 | 0.8873 | ±0.04 |
| | f1-score | **0.9433** | ±0.03 | 0.9139 | ±0.06 | 0.8989 | ±0.05 | 0.8986 | ±0.11 | 0.8890 | ±0.04 |
| | | | | | MFD | | | | | | |
| 0.9 | accuracy | **0.9339** | ±0.11 | 0.8804 | ±0.10 | 0.8881 | ±0.10 | 0.8359 | ±0.09 | 0.9148 | ±0.08 |
| | f1-score | 0.9065 | ±0.15 | 0.8487 | ±0.18 | 0.8737 | ±0.10 | 0.7949 | ±0.09 | **0.9068** | ±0.09 |
| 0.95 | accuracy | **0.9519** | ±0.06 | 0.8894 | ±0.12 | 0.8648 | ±0.11 | 0.8077 | ±0.10 | 0.8915 | ±0.12 |
| | f1-score | **0.9096** | ±0.14 | 0.8673 | ±0.16 | 0.8331 | ±0.12 | 0.7677 | ±0.10 | 0.8764 | ±0.13 |
| | | | | | PTBXL | | | | | | |
| 0.9 | accuracy | **0.7213** | ±0.02 | 0.6938 | ±0.02 | 0.6794 | ±0.04 | 0.6627 | ±0.05 | 0.6733 | ±0.03 |
| | f1-score | **0.5880** | ±0.04 | 0.5862 | ±0.03 | 0.5674 | ±0.05 | 0.5730 | ±0.05 | 0.5687 | ±0.05 |
| 0.95 | accuracy | **0.7014** | ±0.01 | 0.6810 | ±0.04 | 0.6620 | ±0.02 | 0.6594 | ±0.03 | 0.6607 | ±0.03 |
| | f1-score | 0.5701 | ±0.05 | **0.5802** | ±0.04 | 0.5562 | ±0.04 | 0.5792 | ±0.02 | 0.5637 | ±0.04 |
| | | | | | WISDM | | | | | | |
| 0.9 | accuracy | **0.7403** | ±0.11 | 0.7178 | ±0.14 | 0.6885 | ±0.09 | 0.6086 | ±0.15 | 0.6597 | ±0.12 |
| | f1-score | **0.6709** | ±0.15 | 0.6703 | ±0.16 | 0.4315 | ±0.16 | 0.4777 | ±0.16 | 0.6134 | ±0.13 |
| 0.95 | accuracy | 0.6732 | ±0.13 | 0.6594 | ±0.17 | **0.7020** | ±0.06 | 0.6552 | ±0.09 | 0.6462 | ±0.10 |
| | f1-score | 0.6084 | ±0.16 | **0.6204** | ±0.17 | 0.5044 | ±0.10 | 0.5673 | ±0.13 | 0.5970 | ±0.15 |

Table S12: MoSSDA performances of different kernel for mmd loss in RESNET18 backbone. Bold represents best score in the same condition.

| unlabl_ratio | | Scenario ( T to S ) 1_to_2 | 1_to_3 | 2_to_1 | 2_to_3 | 3_to_1 | 3_to_2 |
|---|---|---|---|---|---|---|---|
| 0.7 | accuracy | 0.7263 | 0.7222 | 0.7727 | 0.7440 | 0.7683 | 0.6733 |
| | f1_score | 0.5883 | 0.5517 | 0.6963 | 0.6038 | 0.6796 | 0.5582 |
| | auorc | 0.8194 | 0.8565 | 0.8991 | 0.8384 | 0.8956 | 0.8232 |
| 0.9 | accuracy | 0.6813 | 0.7076 | 0.7098 | 0.7409 | 0.7507 | 0.6973 |
| | f1_score | 0.5503 | 0.5307 | 0.6000 | 0.5840 | 0.6460 | 0.5791 |
| | auorc | 0.8304 | 0.8335 | 0.8546 | 0.8304 | 0.8744 | 0.8335 |
| 0.95 | accuracy | 0.6953 | 0.6837 | 0.7032 | 0.7055 | 0.7135 | 0.7013 |
| | f1_score | 0.5600 | 0.5032 | 0.6052 | 0.5337 | 0.6220 | 0.5865 |
| | auorc | 0.8170 | 0.8143 | 0.8673 | 0.7745 | 0.8552 | 0.8160 |

Table S13: MoSSDA performances are evaluated on PTBXL dataset in all possible domain scenario and 3 fixed unlabeled ratio using RESNET18 as backbone.

| unlab. ratio | | Scenario ( T to S ) 20_to_30 | 23_to_32 | 28_to_4 | 2_to_11 | 33_to_12 | 35_to_31 | 5_to_26 | 6_to_19 | 7_to_18 |
|---|---|---|---|---|---|---|---|---|---|---|
| 0.7 | accuracy | 0.8350 | 0.7826 | 0.8789 | 0.7368 | 0.7931 | 0.8193 | 0.8659 | 0.8788 | 0.7925 |
| | f1_score | 0.7093 | 0.7201 | 0.8268 | 0.5294 | 0.5047 | 0.7461 | 0.8165 | 0.8087 | 0.6563 |
| | auorc | 0.9405 | 0.9171 | 0.9570 | 0.9080 | 0.9730 | 0.9438 | 0.9301 | 0.956 | 0.8204 |
| 0.9 | accuracy | 0.7670 | 0.8116 | 0.8030 | 0.7763 | 0.6437 | 0.8675 | 0.8049 | 0.8106 | 0.7453 |
| | f1_score | 0.6456 | 0.7369 | 0.7648 | 0.6826 | 0.4112 | 0.7335 | 0.6175 | 0.6618 | 0.6305 |
| | auorc | 0.9076 | 0.8767 | 0.9221 | 0.8715 | 0.7011 | 0.9603 | 0.8879 | 0.9437 | 0.7577 |
| 0.95 | accuracy | 0.6893 | 0.6667 | 0.7879 | 0.6447 | 0.6437 | 0.6506 | 0.7683 | 0.7879 | 0.6887 |
| | f1_score | 0.6076 | 0.6461 | 0.7245 | 0.6468 | 0.4841 | 0.3902 | 0.5496 | 0.7253 | 0.5746 |
| | auorc | 0.8940 | 0.8360 | 0.9448 | 0.8077 | 0.8185 | 0.8927 | 0.8738 | 0.9173 | 0.7509 |

Table S14: MoSSDA performances are evaluated on WISDM dataset in randomly fixed 10 domain scenario and 3 fixed unlabeled ratio using TCN as backbone.

| unlbl ratio | metric | AdaMatch Avg. | std. | AdaMatch w/ mixup Avg. | std. | DST Avg. | std. | DST w/ mixup Avg. | std. | UniSSDA Avg. | std. | UniSSDA w/ mixup Avg. | std. |
|---|---|---|---|---|---|---|---|---|---|---|---|---|---|
| | | | | | | EEG | | | | | | | |
| 0.9 | acc | **0.4638** | ±0.06 | 0.4356 | ±0.09 | 0.4572 | ±0.08 | 0.4359 | ±0.10 | 0.4028 | ±0.06 | 0.2913 | ±0.08 |
| | f1 | 0.3755 | ±0.05 | 0.3421 | ±0.08 | **0.3865** | ±0.07 | 0.3469 | ±0.09 | 0.3147 | ±0.07 | 0.1932 | ±0.05 |
| 0.95 | acc | **0.4576** | ±0.06 | 0.4082 | ±0.09 | 0.4522 | ±0.08 | 0.4100 | ±0.10 | 0.3974 | ±0.06 | 0.2851 | ±0.08 |
| | f1 | 0.3718 | ±0.05 | 0.3239 | ±0.08 | **0.3879** | ±0.07 | 0.3348 | ±0.09 | 0.3121 | ±0.07 | 0.1906 | ±0.05 |
| | | | | | | HAR | | | | | | | |
| 0.9 | acc | 0.5071 | ±0.11 | **0.6532** | ±0.08 | 0.5008 | ±0.10 | 0.6302 | ±0.07 | 0.4240 | ±0.06 | 0.5689 | ±0.14 |
| | f1 | 0.3936 | ±0.11 | **0.5979** | ±0.11 | 0.3999 | ±0.09 | 0.5716 | ±0.10 | 0.3096 | ±0.06 | 0.4725 | ±0.17 |
| 0.95 | acc | 0.4947 | ±0.11 | 0.6338 | ±0.12 | 0.4993 | ±0.10 | **0.6342** | ±0.06 | 0.4155 | ±0.07 | 0.5783 | ±0.13 |
| | f1 | 0.3696 | ±0.09 | **0.5739** | ±0.14 | 0.3958 | ±0.08 | 0.5725 | ±0.06 | 0.3022 | ±0.05 | 0.4783 | ±0.15 |
| | | | | | | HHAR | | | | | | | |
| 0.9 | acc | 0.4416 | ±0.16 | **0.4549** | ±0.14 | 0.4525 | ±0.17 | 0.4494 | ±0.14 | 0.4541 | ±0.17 | 0.3834 | ±0.14 |
| | f1 | 0.3973 | ±0.18 | 0.3956 | ±0.15 | **0.4131** | ±0.18 | 0.3957 | ±0.16 | 0.4118 | ±0.19 | 0.3331 | ±0.14 |
| 0.95 | acc | 0.4415 | ±0.16 | **0.4541** | ±0.13 | 0.4499 | ±0.16 | 0.4469 | ±0.14 | 0.4457 | ±0.17 | 0.3786 | ±0.14 |
| | f1 | 0.3957 | ±0.18 | 0.3923 | ±0.15 | **0.4113** | ±0.18 | 0.3934 | ±0.16 | 0.4011 | ±0.18 | 0.3305 | ±0.14 |

Table S15: Comparison the effect of input-level mixup augmentations for the baselines in CNN backbone.

| unlbl ratio | metric | AdaMatch Avg. | std. | AdaMatch w/ mixup Avg. | std. | DST Avg. | std. | DST w/ mixup Avg. | std. | UniSSDA Avg. | std. | UniSSDA w/ mixujp Avg. | std. |
|---|---|---|---|---|---|---|---|---|---|---|---|---|---|
| | | | | | | EEG | | | | | | | |
| 0.9 | acc | **0.4767** | ±0.11 | 0.4437 | ±0.11 | 0.4609 | ±0.13 | 0.4614 | ±0.10 | 0.4160 | ±0.10 | 0.3414 | ±0.12 |
| | f1 | **0.3816** | ±0.10 | 0.3561 | ±0.10 | 0.3776 | ±0.12 | 0.3670 | ±0.09 | 0.3223 | ±0.10 | 0.2363 | ±0.09 |
| 0.95 | acc | **0.4684** | ±0.11 | 0.4239 | ±0.11 | 0.4558 | ±0.12 | 0.4470 | ±0.10 | 0.4046 | ±0.09 | 0.3400 | ±0.12 |
| | f1 | **0.3806** | ±0.11 | 0.3455 | ±0.10 | 0.3806 | ±0.11 | 0.3647 | ±0.09 | 0.3121 | ±0.10 | 0.2338 | ±0.09 |
| | | | | | | HAR | | | | | | | |
| 0.9 | acc | **0.6001** | ±0.09 | 0.5498 | ±0.07 | 0.5880 | ±0.06 | 0.5690 | ±0.07 | 0.5191 | ±0.04 | 0.4237 | ±0.12 |
| | f1 | **0.5333** | ±0.11 | 0.4965 | ±0.08 | 0.4946 | ±0.07 | 0.5040 | ±0.09 | 0.3862 | ±0.04 | 0.2699 | ±0.12 |
| 0.95 | acc | **0.6071** | ±0.09 | 0.5508 | ±0.09 | 0.5871 | ±0.07 | 0.5712 | ±0.06 | 0.5221 | ±0.04 | 0.4239 | ±0.11 |
| | f1 | **0.5332** | ±0.11 | 0.5018 | ±0.09 | 0.4912 | ±0.08 | 0.5072 | ±0.08 | 0.3956 | ±0.05 | 0.2856 | ±0.10 |
| | | | | | | HHAR | | | | | | | |
| 0.9 | acc | 0.5093 | ±0.16 | 0.4693 | ±0.16 | **0.5159** | ±0.18 | 0.4748 | ±0.14 | 0.4987 | ±0.16 | 0.4685 | ±0.12 |
| | f1 | 0.4935 | ±0.18 | 0.4507 | ±0.17 | **0.5071** | ±0.21 | 0.4591 | ±0.16 | 0.4830 | ±0.18 | 0.4236 | ±0.15 |
| 0.95 | acc | **0.5096** | ±0.16 | 0.4715 | ±0.15 | 0.5043 | ±0.18 | 0.4739 | ±0.14 | 0.4998 | ±0.16 | 0.4749 | ±0.13 |
| | f1 | **0.4915** | ±0.18 | 0.4508 | ±0.17 | 0.4900 | ±0.21 | 0.4560 | ±0.15 | 0.4816 | ±0.18 | 0.4268 | ±0.15 |

Table S16: Comparison the effect of input-level mixup augmentations for the baselines in RESNET18 backbone.

yields better performance than its mixup counterpart. These results suggest that the effectiveness of input-level mixup is highly sensitive to both the dataset and the choice of backbone architecture.

