# OpenReview forum: "MoSSDA: A Semi-Supervised Domain Adaptation Framework for Multivariate Time-Series Classification using Momentum Encoder"
_ICLR.cc/2026/Conference — Submitted to ICLR 2026_

### Official Review · Reviewer_ALKM · 2025-10-27

**Soundness:** 2
**Presentation:** 2
**Contribution:** 2
**Rating:** 4
**Confidence:** 3

**Summary:**

Thank you for the opportunity to review this paper. This paper proposes MoSSDA, a two-stage semi-supervised domain adaptation (SSDA) framework for multivariate time-series classification that tackles domain shift when only a few target labels are available. In the first stage, it learns domain-invariant and class-discriminative representations using a domain-invariant encoder with Maximum Mean Discrepancy (MMD) loss and a positive contrastive module combining mixup and a momentum encoder for stable, consistent feature learning. In the second stage, a classifier is trained on the frozen features to prevent conflicting objectives. Evaluated on six benchmark datasets and multiple backbones, MoSSDA achieves state-of-the-art accuracy and robustness even with 95% unlabeled target data, outperforming existing SSDA methods. Ablation studies show that the contrastive module and two-stage design are key to its success.

**Strengths:**

1. The studied problem is important.
2. The proposed framework seems reasonable.
3. Extensive experiments are conducted to demonstrate the effectiveness of the proposed method.

**Weaknesses:**

1. The presentations and novelty can be illustrated better.
2. Some literatures are missing.
3. Comparison methods may not be up-to-date.

**Questions:**

This paper tackles an important problem in semi-supervised domain adaptation for time-series classification, and the proposed framework seems promising based on the reported results. However, several aspects could be improved to make the contribution clearer and more convincing.

1. Some parts of the introduction and methodology are difficult to follow. It’s not clearly explained why domain shift and violations of the i.i.d. assumption degrade performance, or what specific challenge this work aims to solve beyond what has already been addressed in prior domain adaptation research. Since domain shift is a well-studied issue, readers may find it hard to understand what is truly new here. The proposed “domain-invariant” framework sounds similar to many existing methods, and the paper should emphasize more clearly what the technical novelty is. It would also help to discuss whether the approach could generalize beyond classification to other time-series tasks such as forecasting or anomaly detection.

2. The related-work section misses several recent and relevant papers. In particular, there has been active research on source-free and multi-source domain adaptation for time-series data. Examples include:

   Time-series domain adaptation via sparse associative structure alignment: Learning invariance and variance (Neural Networks, 2024)

   Source-free domain adaptation with temporal imputation for time series data (KDD 2023)

   POND: Multi-source time series domain adaptation with information-aware prompt tuning (KDD 2024)

     It would strengthen the paper if the authors could discuss these works and explain whether they could be applied to the problem setting considered here. Ideally, a comparison with some of these recent methods would also be discussed.

3. Most of the comparison methods are from before 2023. Given how quickly the area is evolving, it would be good to include more recent state-of-the-art baselines to better assess the performance of the proposed approach.

**Details Of Ethics Concerns:**

No ethics concerns.

---

> ### Author Response · Authors · 2025-11-20
>
> We appreciate your attention to detail and the constructive nature of your comments.
> Below, we respond to each comment in turn and describe the corresponding revisions made to the paper.
> 1. Thank you for raising this important concern about clarity. We agree that additional explanation is needed in the introduction and methodology to more clearly motivate our problem setting and contributions. To address this, we will revise Section 1 to provide a more detailed explanation of the i.i.d. assumption: standard supervised learning assumes that the training and test samples come from the same distribution, but domain shift in real-world time series often violates this assumption, degrading model performance. In Section 3, we will clarify that our key technical novelty is eliminating conflicting gradients by explicitly separating representation learning (stage 1) and classifier training (stage 2)—unlike previous works that jointly optimize domain alignment and class discrimination, which can result in instability. Our MoSSDA framework is the first, to our knowledge, to combine feature-space Mixup with a two-stage learning process and momentum encoding for time-series SSDA. Finally, we will more explicitly discuss the applicability of our approach beyond classification and outline its potential for extending to other tasks such as forecasting and anomaly detection, thereby clarifying its broader relevance.
> 2. You are absolutely correct that discussions of source-free and multi-source domain adaptation for time series are highly relevant. We will substantially expand Section 2 to include these papers, discuss their approaches, and clarify how they relate to and differ from our problem setting. Where possible, we will also aim to incorporate comparisons or references to these methods to strengthen the context and contributions of our work.
> 3.  Our baseline methods were primarily chosen from well-established semi-supervised domain adaptation approaches commonly applied and adapted for time series data. We acknowledge that some recent state-of-the-art methods published after 2023 may not have been included. We will make efforts to update our experimental comparisons to incorporate more recent baselines where applicable to provide a more comprehensive assessment of our proposed approach.

---

> ### Author Response · Authors · 2025-11-27
>
> In the revised manuscript, we have expanded the Introduction to provide a clearer explanation of how violations of the i.i.d. assumption under domain shift degrade performance and to more explicitly highlight the technical novelty and specific challenge addressed in this work.
> In addition, we updated the Section 2.1 to incorporate a discussion of the recent methods.

---

### Official Review · Reviewer_ae3v · 2025-10-30

**Soundness:** 4
**Presentation:** 3
**Contribution:** 3
**Rating:** 6
**Confidence:** 3

**Summary:**

This work introduces MoSSDA, a method for domain adaptation of time series data in the semi-supervised setting. The authors use various methods including MoCo-like contrastive learning, MMD loss, and mix-up to boost performance on this task for downstream classification. They show that their method significantly outperforms baselines in a variety of settings, and through ablations, they establish the importance of each component of their model. Overall, this work introduces a novel method for domain adaptation in time series that can be applied on a variety of downstream datasets with many types of architectures.

**Strengths:**

1. The components of this method are well-motivated based on the problem the authors present. Each component seems to have a separate but important place that meets the challenges of domain adaptation for time series.
2. The benchmarking results are very impressive compared to baselines, showing substantial improvements in this setting across multiple datasets.
3. This method demonstrates how a combination of methods that have been tested extensively in other research domains, such as mix-up, MoCo, and MMD loss, can result in a method that’s performative in time series domain adaptation.
4. The appendix contains a plethora of experiments on their method, and extensive ablations are conducted in order to understand each model component and how the approach performs on different base architectures. This helps greatly in understanding the contributions of this method.

**Weaknesses:**

1. The proposed method is only applicable in somewhat limited settings for domain adaptation. The motivation for self-supervised DA is clear, but the authors only consider the case where label spaces of both source and target domain are known to be entirely overlapping. This is a major weakness as prevailing methods in domain adaptation consider the unsupervised domain adaptation setting where some labels in the target domain might not be known when training the model.
2. The method uses MMD as the main domain adaptation loss. Follow-ups to the MMD have been proposed such as the Sinkhorn divergence (see RAINCOAT, He et al., 2023).
3. In terms of novelty, this method is more of a combination of various methods from other fields rather than a central new method for domain adaptation. The authors motivate the transfer of these components well, but the novelty comes more from the combination of the methods rather than a central new method.
4. The mix-up process seems to be quite restrained by the availability of same-class target samples at training. Since mix-up is only performed between samples in the same class to enable the supervised contrastive learning, this requires that there are a sufficient number of samples in the training set of the target domain such that you can combine them in various ways to get mixed-up samples.

**Questions:**

1. Could this method easily be extended to account for non-identical label sets in source and target domains?
2. The authors claim that certain time series augmentation techniques were needed for the baseline methods in this work, but they claim that such augmentations introduce biases and implausible transformations into the data. Did they test which type of augmentations change the performance of the baseline methods? This is not required on a large scale but should at least be commented on.
3. In the appendix, results across different unlabeled ratios and architectures show that removing the contrastive loss results in near-identical classification metrics for each dataset (Tables S4, S5, S6, S7). Why is this? Does this amount to the model assigning one label to all samples? Comparing this to a naive majority-class classifier could help answer this question. If so, is the method’s central contribution the supervised contrastive learning?
4. Is the momentum encoder needed here? What happens if authors replace the stop-grad and momentum encoder with a regular contrastive learning setup where the encoder is used on both sides and optimized for all samples?

**Details Of Ethics Concerns:**

Codebase submission is a Google Drive link and is not anonymized. Also no LLM statement is included (but ok since this isn't necessarily required if authors didn't use them).

---

> ### Author Response · Authors · 2025-11-20
>
> Your feedback has been instrumental in clarifying and strengthening our arguments.
> We kindly confirm that the submitted codebase has been thoroughly anonymized. Additionally, we assure you that the LLM statement was indeed included at the time of submission. Thank you for your careful review and for allowing us to clarify these points.
> Below, we respond to each comment in turn and describe the corresponding revisions made to the paper.
>
> 1. Extending our method to handle non-identical label sets between source and target domains is an interesting and important direction. We believe that incorporating few-shot or zero-shot learning techniques could effectively address this challenge. Exploring this extension is part of our future work to broaden the applicability of our framework.
> 2. We agree that an ablation study on augmentation types for baseline methods is necessary to understand their varying impacts. We will add an appendix section titled "Augmentation Ablation for Baselines," where we will compare different augmentation variants used in baselines such as AdaMatch and PAC, including input-level Mixup, additive noise, and their combinations. This controlled comparison will help clarify which augmentation strategies positively or negatively influence results, strengthening the rigor of our experimental evaluations.
> 3. Thank you for your insightful observation. We acknowledge that our method combines established techniques from different fields, including supervised contrastive learning and Mixup. However, the novelty lies in the carefully designed two-stage framework that integrates these components to effectively address domain-adaptation challenges in time-series data. The first stage learns a domain-invariant, class-discriminative representation via supervised contrastive learning enhanced with cross-domain Mixup, thereby expanding the positive sample space and improving feature smoothness. The second stage trains a classifier on these robust features. This synergy, along with the separation of gradient flows, is central to the improved performance observed.
> Regarding the concern that near-identical metrics without contrastive loss may indicate label collapse, we will include additional experiments in the Appendix. These will show detailed verification of label collapse, including confusion matrices across all scenarios, and an ablation study comparing our MoSSDA framework with standard contrastive learning. We will clarify these points in the paper to emphasize the holistic and practical contribution of our approach beyond simply combining existing methods.
> 4. The momentum encoder plays a crucial role in maintaining a consistent and stable representation of features used as keys in contrastive learning. By updating the key encoder weights with a momentum mechanism, the model avoids rapid changes that can destabilize training and degrade performance. Replacing the momentum encoder and stop-gradient mechanism with a regular contrastive learning setup—where both encoders are trained end-to-end—can result in less stable training and poorer representation quality. We plan to conduct this important experiment comparing the full MoSSDA framework with and without the momentum encoder (i.g, regular contrastive learning) and will include the results in the Appendix to empirically demonstrate its impact.

---

> > ### Comment · Reviewer_ae3v · 2025-11-20
> >
> > Thank you for your responses! I think your provided points make total sense, I appreciate you taking the time to carefully respond. You mention several new experiments in your comment, but do you have any preliminary results on these? In particular, preliminary results for points 3 and 4 that you mention are in particular important for understanding the benefits of your proposed method. I'll need to see these before I can consider changing my score.

---

> ### Author Response · Authors · 2025-11-27
>
> We appreciate your follow-up and agree that preliminary results are important for assesing the benefits of our method.
> For 3, we have added the corresponding results to the ablation study. Specifically, we updated Table 3 and Section 4.3 Ablation Study and revised Talbes S4, S5, S6, and S7 to more clearly show the effect of remosvng the contrastive loss.
> For 4, we have likewise included new ablation results in the same section, and we additionally report a comparison across different momentum settings for the momentum encoder in Table S10 in Appendix A.7.1.
> Furthermore, for 2, we are currently running additional experiments on the effect of changing augmentations for the baselines. We plan to incorporate these results into the revised manuscript by December 3.

---

### Official Review · Reviewer_pSYM · 2025-10-31

**Soundness:** 3
**Presentation:** 3
**Contribution:** 2
**Rating:** 4
**Confidence:** 4

**Summary:**

This paper a two-stage decoupled training framework to handle the SSDA problem in time series.

**Strengths:**

Strengths:
1. The two-stage decoupled training framework proposed in this paper is an effective strategy for handling complex multi-objective optimization problems, avoiding gradient conflicts between feature extraction and classifier training, which makes significant contributions to the optimization stability of existing SSDA methods.
2. The paper achieves a clever application of Mixup in SSDA by applying it in the feature space rather than the input space, enhancing class discriminability with limited target domain label data while avoiding the potential damage to critical temporal sequential features that traditional input-level augmentation might cause, demonstrating a profound understanding of time series data characteristics.
3. Modular Design and Robustness: The modules of MoSSDA are well-designed with high coupling and each performs its specific function, respectively solving domain invariance, class discriminability, and feature consistency issues. Through extensive experiments, the effectiveness of MoSSDA has been validated.

**Weaknesses:**

Weaknesses：
1. The original theoretical foundation of Mixup lies in encouraging the model to perform linear interpolation between training samples to enhance the model's generalization ability. This paper applies it in the feature space combined with contrastive learning. Is the theoretical effectiveness of this feature space Mixup still equivalent to that of input space Mixup? For the virtual positive samples generated by feature space Mixup in the feature space of time series, can they truly represent meaningful, intra-class compact features? Or, does this linear interpolation introduce unnecessary noise or bias in the nonlinear feature space, especially when the target domain labels are extremely sparse? This point may require some analysis by the authors. Also, the usage of mixup in feature space is not new to the community.
2. The paper employs a linear kernel in the MMD loss and mentions that the framework allows for replacing it with other kernels such as RBF. However, the performance of MMD is highly sensitive to the choice of kernel function. For complex multivariate time series features, the measurement capability of a linear kernel may be insufficient to capture domain shifts in high-dimensional nonlinear feature spaces. The authors should experimentally compare the performance of the linear kernel with other more complex kernel functions, such as RBF or kernels specifically designed for time series, in the MMD loss. If experiments have been conducted, it should be clarified whether, after comparison, the linear kernel achieves SOTA performance in the feature space of this paper.
3. The overall loss function of MoSSDA includes two weight hyperparameters, and this multi-objective optimization problem may be more sensitive to hyperparameter settings. The paper lacks sufficient analysis of this issue and needs to further demonstrate the robustness of these two weights under different datasets and label ratios.
4. The momentum encoder aims to maintain feature consistency. However, in the SSDA setting, the target domain data may contain domain-specific features, and the momentum update mechanism may excessively smooth out the unique and valuable fine-grained features specific to the target domain. The ablation study section of the paper does not demonstrate the impact of the momentum encoder on target domain-specific features. It is recommended that the paper supplement this part to make the experimental setup more comprehensive.
5. The framework proposed in the paper is based on offline two-stage training. In some industrial application scenarios, such as sensor monitoring and equipment fault diagnosis, target domain data is continuously streamed, and the model may need to possess online or incremental adaptation capabilities. Some arguments about how the MoSSDA framework adapts to data streams online can be supplemented.
6. As SSDA is a very old area in machine learning. It is important to clarify what are the special designs in time series domain. Can the proposed method be used in general SSDA setting?

**Questions:**

As in Weaknesses

---

> ### Author Response · Authors · 2025-11-20
>
> Thank you for your careful reading and constructive criticism. We have carefully considered all of your points.
> Below, we respond to each comment in turn and describe the corresponding revision made to the paper.
>
> 1. Thank you for raising this important point. Mixup indeed enhances generalization by enabling richer representations through linear interpolation. In our work, we apply Mixup across same-class samples from different domains to leverage complementary information, effectively utilizing scarce target-domain labels and encouraging domain-generalizable class features.
> We believe feature-space Mixup is more appropriate than input-level Mixup for time series data, as it better captures high-level semantic information without disturbing raw temporal signals. While we acknowledge potential concerns about noise or bias in nonlinear feature spaces, our empirical results demonstrate improved performance. Although feature-space Mixup is well known, our focus on cross-domain time-series data with label scarcity offers a novel and practical contribution.
> 2. This is an excellent point that deserves thorough investigation. We acknowledge this as a critical technical gap. We add these important results to the Appendix. We will compare MMD performance across kernels (two different RBF kernels and two different polynomial (rank=2 and 3) kernels) on representative datasets.
> 3. Excellent observation. We will add a new figure and analysis to the Appendix. Hyperparameter greedy search results on representative datasets : λ_mmd, λ_ctr ∈ {0.3, 0.5, 0.7}
> 4. Thank you for raising this nuanced concern about the balance between maintaining feature consistency and preserving target-domain-specific features. The momentum encoder's update mechanism could potentially smooth out fine-grained, domain-specific features, which are particularly valuable in the SSDA setting with sparse target-domain labels. To address this, we plan to conduct additional experiments to analyze the impact of momentum coefficients on target-domain features. We will include a detailed examination of how the momentum encoder affects the preservation of these domain-specific features in our ablation study. We believe such analyses will provide a clearer understanding of the tension between consistency and specificity in our framework.
> 5. Our current work assumes that a substantial labeled source-domain dataset is available, along with some annotated target-domain data. For example, in smartwatches, while we have collected data for model development, adapting to new users typically involves an initial calibration phase where label information can be obtained. Based on this understanding, our proposed method focuses on this practical scenario. From your insightful suggestion, we recognize the importance of extending MoSSDA's applicability to online or incremental adaptation settings where target data streams continuously. We plan to explore this direction as future work to enhance the framework's scalability and real-world relevance.
> 6. While SSDA has a long history in machine learning. Its application in time series domains presents unique challenges and opportunities. Complex time series signals frequently occur in critical real-world applications such as healthcare, where monitoring vital signs from wearable devices requires adaptation to individual patient variability, and industrial process monitoring, for example, in manufacturing or equipment fault diagnosis, where multi-sensor temporal data streams must be analyzed for timely anomaly detection. In these domains, time series data is often high-dimensional, noisy, and exhibits domain shifts that standard SSDA methods may not fully address.
> Our proposed approach incorporates designs tailored to capture temporal dynamics and domain-specific patterns effectively, which are crucial to robust performance in these settings. While the method is motivated by time-series applications, the framework can be adapted to more general SSDA scenarios. By showcasing use cases from healthcare and industrial monitoring, we demonstrate the practical necessity and benefits of specialized SSDA techniques for time series data.

---

> > ### Comment · Reviewer_pSYM · 2025-11-27
> >
> > Thank you for your response.
> >
> > However, rather than alleviating my concerns, your rebuttal has unfortunately intensified them. My primary skepticism revolves around the lack of methodological novelty and, specifically, the absence of clarification regarding specialized designs for the time-series domain. Your response did not address these core issues with any substantive details.
> >
> > My assessment is based on the following points:
> >
> > 1. Lack of Innovation in Core Components: The fundamental elements of your framework (MMD, Feature Mixup, Momentum Encoder, and Contrastive Learning) are all established techniques in transfer and semi-supervised learning that were pioneered years ago. For instance:
> >
> > MMD for domain adaptation dates back to the 2014 DDC paper [1].
> >
> > Feature Mixup was systematically explored for transfer learning in the 2021 SMILE method [2].
> >
> > The Momentum Encoder paradigm was introduced by the Mean Teacher method in 2017 [3].
> > Consequently, this paper does not introduce novelty at the level of methodology but may, at best, represent an application of existing techniques to time-series data.
> >
> > 2. The Inquiry on "Special Design" Was Evaded: My original question explicitly asked for the specialized designs for time-series. Your response vaguely claims to "incorporate designs tailored to capture temporal dynamics" but fails to specify what these designs are. What are the specific architectural choices, loss functions, or algorithmic steps that constitute this design? How do they fundamentally differ from applying generic SSDA methods (e.g., using MMD or Mixup directly on image features)? Without these details, the work appears to be an application study rather than a methodological contribution.
> >
> > 3. This Omission Mirrors the Manuscript: This lack of critical detail is also reflected in the paper itself. For example, the introduction spends considerable space (35 lines) discussing the importance of SSDA, yet devotes only a minimal portion (5 lines) to describing the proposed method's core design, without clarifying what makes it uniquely suited for time-series compared to existing approaches.
> >
> > Therefore, I must reiterate my questions and insist on concrete, technical answers:
> >
> > Please explicitly identify which components of your model architecture, loss function, or training algorithm are specifically designed to handle temporal dynamics. Provide precise details.
> >
> > How do these designs, in principle, make your method superior or different from directly applying existing generic SSDA methods (e.g., those based on MMD or Mixup) to time-series features?
> >
> > Specifically, what essential modifications would be required to apply your framework to an image-based SSDA task? This will help clarify the specificity of your approach.
> >
> > If the authors cannot clearly and concretely elucidate the points above, it will confirm that this paper lacks sufficient methodological innovation, and I will consequently maintain my recommendation for rejection.
> >
> > [1] Deep domain confusion: Maximizing for domain invariance. 2014.
> > [2] Smile: Self-distilled mixup for efficient transfer learning. 2021.
> > [3] Mean teachers are better role models: Weight-averaged consistency targets improve semi-supervised deep learning results. 2017.

---

> ### Author Response · Authors · 2025-11-27
>
> In the revised manuscript, we have made the following additions and clarifications.
> We conducted an MMD kernel sensitivity analysis, now presented in Appendix A.7.2, with detailed results reported in Table S11.
> We added experiments on the loss-weight hyperparamters. The results of different hyperparameter combinations are provided in Appendix A.7.1, and summarized in Tables S8 and S9.
> To address the concern about the momentum encoder, we extended the ablation study by including results without the momentum encoder, and we additionally report a sensitivity analysis over the momentum parameter in Table S10 in Appendix A.7.1.
> Finally, we expanded the Conclusion section to discuss potential limitations and future work.

---

> ### Author Response · Authors · 2025-11-28
>
> Thank you again for your detailed and candid feedback. We fully agree that MMD, Mixup, momentum encoders, and contrastive learning are established techniques. Our contribution is not to introduce new primitives, but to reorganize them into a time‑series–oriented SSDA framework that directly addresses failure modes we observed when generic SSDA is applied to temporal data.
>
> Our “special designs” are three concrete changes to address these issues.
>
> (1) Decoupled two‑stage training (training algorithm + loss design).
> In Stage 1, a domain‑invariant encoder is trained on source + labeled and unlabeled target using a representation loss composed of MMD and supervised contrastive terms on mixed features. In Stage 2, the encoder is frozen, and only the classifier is trained with cross‑entropy on the labeled source and target. For time‑series, joint optimization of (CE + MMD + contrastive) produced strong gradient conflicts and degraded temporal features, while this two‑stage schedule stabilized training and 17.4% acc, 19.3% F1 gain. Thus, the novelty is in how we schedule and isolate alignment/contrastive signals for noisy sequential data, not in simply using MMD.
>
> (2) Feature‑space Mixup (where and how Mixup is applied).
> Existing time‑series DA methods, such as CoTMix, perform Mixup directly on the time axis. In our experiments, raw temporal mixing or strong time‑warping/permutation distorted waveform morphology and hurt SSDA performance. We therefore avoid manipulating the raw temporal axis and instead construct mixed features from same‑class features (often from different domains), using them as additional positives in the contrastive loss. This preserves the temporal structure learned by the encoder while still providing Mixup‑style regularization and inter‑domain smoothing. Removing this feature‑space Mixup, while keeping the rest of the framework unchanged, consistently decreased performance across datasets, indicating that this time‑series–motivated location of Mixup is essential and different from applying Mixup directly to image or time‑series inputs.
>
> (3) Momentum encoder as a temporal pattern anchor.
> We maintain an online encoder and a momentum encoder updated via an exponential moving average of the online parameters. While momentum encoders in Mean‑Teacher‑style methods act as generic stabilizers, in our time‑series SSDA setting, we specifically use this mechanism to anchor recurring temporal motifs across domains and training iterations, especially for long, noisy sequences where representations easily drift. In ablation, removing the momentum encoder systematically reduced performance and produced less coherent cross‑domain clusters in the embedding space, supporting our interpretation that it acts as a temporal anchor under domain shift, rather than just another semi‑supervised regularizer.
>
> These three elements—two‑stage optimization, feature‑space Mixup, and a momentum‑based temporal anchor—are implemented on standard 1D CNN/ResNet/TCN backbones. We deliberately avoid designing a bespoke time‑series architecture so that the time‑series specialization lies in the training strategy and loss design, not in the encoder itself.
>
> Regarding your question on how this differs from directly applying generic SSDA (e.g., MMD‑ or Mixup‑based methods) to time‑series features: the difference is not in the names of the components, but in where and how they are used. Generic SSDA typically (i) trains encoder and classifier jointly with a single combined loss and (ii) relies on input‑space augmentations or Mixup. By contrast, our method (a) separates representation learning from classifier training to mitigate gradient conflicts that are particularly severe under temporal noise, (b) avoids manipulating the raw temporal axis and performs Mixup only after temporal encoding, and (c) uses a momentum encoder to stabilize cross‑domain temporal patterns when strong input augmentations are intentionally avoided. In our experiments, simply plugging MMD, input‑space Mixup, and a momentum encoder into a standard one‑stage SSDA pipeline for time‑series did not match the performance of our framework; the gains are tied to this time‑series–oriented restructuring.
>
> Finally, to clarify the specificity to time‑series, if we were to adapt our framework to an image‑based SSDA task, several components would naturally change: 1D encoders would be replaced by 2D CNNs, and temporal receptive fields would no longer be relevant; aggressive input‑level augmentations and input‑space Mixup (which we avoid for temporal data) would again be appropriate and beneficial; and the reliance on a two‑stage schedule and on the momentum encoder as a temporal anchor would likely be weaker, because joint optimization is typically more stable for images. We will make these motivations and design choices more explicit in the revised manuscript to clarify the time‑series–specific methodological contribution.

---

### Official Review · Reviewer_BkaN · 2025-11-01

**Soundness:** 3
**Presentation:** 3
**Contribution:** 2
**Rating:** 4
**Confidence:** 3

**Summary:**

The paper proposes MoSSDA (Momentum encoder-utilized Semi-Supervised Domain Adaptation), a two-stage framework for addressing domain shift in multivariate time-series classification when limited labeled target data is available. The key innovation is combining a domain-invariant encoder, a mixup-enhanced positive contrastive module with momentum encoding, and a decoupled training strategy to learn robust representations without requiring input-level augmentations.

**Strengths:**

1. The paper is well-motivated and structured
2. Comprehensive experiments: The evaluation spans 6 diverse datasets, 3 backbone architectures, and multiple unlabeled ratios (0.7, 0.9, 0.95), demonstrating consistent improvements over strong baselines. Authors also provide a ablation study, which clearly demonstrate the contribution of each component, with the positive contrastive module showing the most significant impact.

**Weaknesses:**

Limited novelty. The combination of MMD loss for domain alignment, mixup-enhanced contrastive learning, and momentum encoding is creative and well-justified for time-series data where traditional augmentations can disrupt temporal dependencies. While the combination is novel, the individual components (MMD loss, contrastive learning, momentum encoding) are well-established techniques. The main contribution is their integration for time-series SSDA.

**Questions:**

1. The claim about creating "rich representations without input-level augmentations" is misleading since mixup is itself a form of augmentation, just in feature space rather than input space.
2. The paper doesn't discuss potential failure modes or limitations of the approach.
3. The MMD loss uses a linear kernel by default, which may not be sufficient for complex distribution differences. why this simple kernel would work for all datasets.

---

> ### Author Response · Authors · 2025-11-20
>
> Thank you very much for your thoughtful review and valuable feedback on our study.
> Below, we respond to each comment in turn and describe the corresponding revisions made to the paper.
>
> 1. We appreciate this clarification. You are correct that our phrasing may have been imprecise. To better articulate our intention, we propose revising the abstract and introduction as follows: " … without input-level time-series augmentations that disrupt temporal structure, instead employing feature-space mixup to preserve temporal dependencies while enhancing class discriminability."
> Our distinction is motivated by a critical difference. Traditional input-level augmentations (e.g., rotation, cropping, permutation, random erasing, time warping) can fundamentally alter or destroy the temporal ordering and dependencies that are semantically meaningful in time series. In contrast, feature-space Mixup operates in the learned representation space where temporal structure has already been implicitly encoded by the encoder, thus avoiding direct manipulation of temporal relationships. We will add this clarification to Section 3.4 (Positive Contrastive Module).
>
> 2. You are absolutely right. In the revised version, we will add a discussion about potential failure modes or limitations that comprehensively address severe class imbalance, backbone architecture dependency, and incremental adaptation.
>
> 3. This is an excellent point that deserves thorough investigation. We propose conducting and including the following experiments.
> We will compare MMD performance across kernels (two different RBF kernels and two different polynomial (rank=2 and 3) kernels) on representative datasets.

---

> ### Author Response · Authors · 2025-11-27
>
> In the revised version, we have addressed all three points as follows.
> First, we clrified our claim regarding "rich representations without input-level augmentations" in the Introduction and Section 3.4 to better articulate our intention and explicitly acknowledge the role of mixup as feature-space augmentation.
> Second, we added a discussion of potential failure modes and limitations in the Conclusion.
> Third, we conducted an MMD kernel sensitivity analysis, which is now reprted in Appendix A.7.2, with the corresponding experimental results presented in Table S11.

---

### Meta-Review · Area_Chair_uNgH · 2025-12-25

**Summary:**

The submission introduces MoSSDA, a semi-supervised domain adaptation framework specifically designed for multivariate time-series classification. The authors propose a two-stage training paradigm that leverages a momentum encoder and feature-space Mixup to align domains. The core motivation is to learn domain-invariant representations while preserving temporal dependencies, avoiding the potential signal disruption caused by standard input-level augmentations.

**Reviewer Concerns:**

The authors made a commendable effort during the rebuttal phase, effectively addressing several empirical questions regarding the sensitivity of MMD kernels, the discussion of failure modes, and the inclusion of missing literature on source-free and multi-source adaptation. These revisions improved the completeness of the manuscript.

However, the primary concern regarding technical novelty remains unresolved. Multiple reviewers (notably pSYM and BkaN) argue that the proposed framework represents an incremental engineering combination of well-established techniques—namely MMD, Mixup, and Momentum Contrast—rather than a fundamental algorithmic innovation. The authors' defense of "feature-space Mixup" as a specialized design for time-series was met with skepticism, as it is viewed as a standard application of generic representation learning methods. The consensus is that while the method is sound, it lacks the distinct methodological contribution required for ICLR.

**Reviewer Scores:**

Reviewer ae3v (Score 6) reacted positively to the additional ablation studies and is likely to maintain their support. Conversely, Reviewer pSYM (Score 4) explicitly stated that the rebuttal intensified their concerns about the lack of specific time-series design elements and is expected to maintain or lower their score. Reviewers BkaN and ALKM (Scores 4), while acknowledging the improved presentation, remain unconvinced by the magnitude of the contribution and are unlikely to move their scores significantly above the acceptance bar.

---

### Decision · Program_Chairs · 2026-01-26

Reject